# HippoRAG: Neurobiologically Inspired Long-Term Memory for Large Language Models

**Bernal Jiménez Gutiérrez**
The Ohio State University

**Yiheng Shu**
The Ohio State University

**Yu Gu**
The Ohio State University

**Michihiro Yasunaga**
Stanford University

**Yu Su**
The Ohio State University

## Abstract

In order to thrive in hostile and ever-changing natural environments, mammalian brains evolved to store large amounts of knowledge about the world and continually integrate new information while avoiding catastrophic forgetting. Despite their impressive accomplishments, large language models (LLMs), even with retrieval-augmented generation (RAG), still struggle to efficiently and effectively integrate a large amount of new experiences after pre-training. In this work, we introduce HippoRAG, a novel retrieval framework inspired by the hippocampal indexing theory of human long-term memory to enable deeper and more efficient knowledge integration over new experiences. HippoRAG synergistically orchestrates LLMs, knowledge graphs, and the Personalized PageRank algorithm to mimic the different roles of neocortex and hippocampus in human memory. We compare HippoRAG with existing RAG methods on multi-hop question answering (QA) and show that our method outperforms the state-of-the-art methods remarkably, by up to 20%. Single-step retrieval with HippoRAG achieves comparable or better performance than iterative retrieval like IRCoT while being 10-20 times cheaper and 6-13 times faster, and integrating HippoRAG into IRCoT brings further substantial gains. Finally, we show that our method can tackle new types of scenarios that are out of reach of existing methods.[1]

## 1 Introduction

Millions of years of evolution have led mammalian brains to develop the crucial ability to store large amounts of world knowledge and continuously integrate new experiences without losing previous ones. This exceptional long-term memory system eventually allows us humans to keep vast stores of continuously updating knowledge that forms the basis of our reasoning and decision making [19].

Despite the progress of large language models (LLMs) in recent years, such a continuously updating long-term memory is still conspicuously absent from current AI systems. Due in part to its ease of use and the limitations of other techniques such as model editing [46], retrieval-augmented generation (RAG) has become the *de facto* solution for long-term memory in LLMs, allowing users to present new knowledge to a static model [36, 42, 66, 87].

However, current RAG methods are still unable to help LLMs perform tasks that require integrating new knowledge across passage boundaries since each new passage is encoded in isolation. Many important real-world tasks, such as scientific literature review, legal case briefing, and medical diagnosis, require knowledge integration across passages or documents. Although less complex,

---

[1]Code and data are available at `https://github.com/OSU-NLP-Group/HippoRAG`.

38th Conference on Neural Information Processing Systems (NeurIPS 2024).

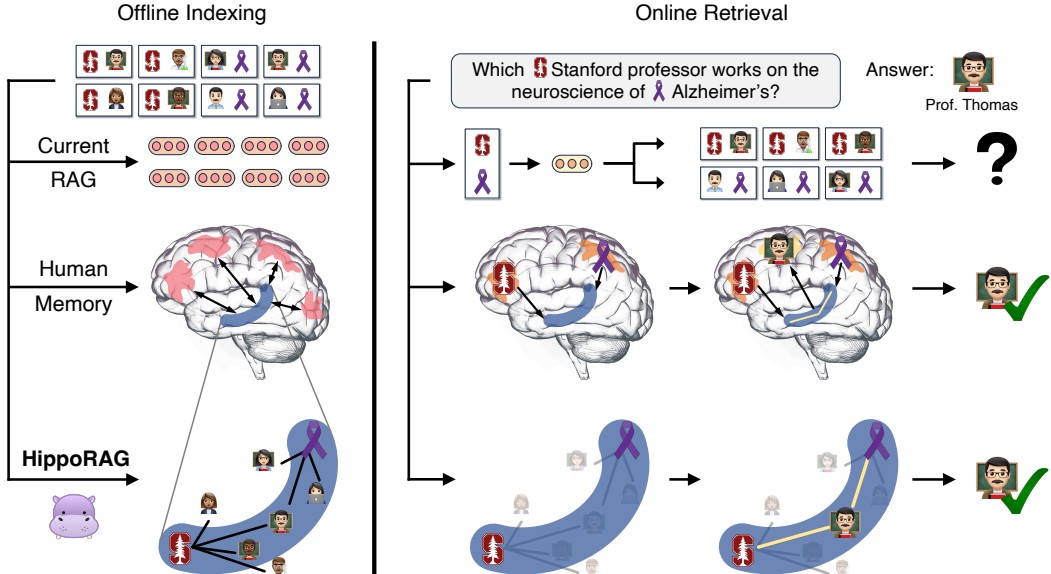

Figure 1: **Knowledge Integration & RAG.** Tasks that require knowledge integration are particularly challenging for current RAG systems. In the above example, we want to find a *Stanford* professor that does *Alzheimer's* research from a pool of passages describing potentially thousands *Stanford* professors and *Alzheimer's* researchers. Since current methods encode passages in isolation, they would struggle to identify *Prof. Thomas* unless a passage mentions both characteristics at once. In contrast, most people familiar with this professor would remember him quickly due to our brain's associative memory capabilities, thought to be driven by the index structure depicted in the C-shaped hippocampus above (in blue). Inspired by this mechanism, **HippoRAG** allows LLMs to build and leverage a similar graph of associations to tackle knowledge integration tasks.

standard multi-hop question answering (QA) also requires integrating information between passages in a retrieval corpus. In order to solve such tasks, current RAG systems resort to using multiple retrieval and LLM generation steps iteratively to join disparate passages [64, 78]. Nevertheless, even perfectly executed multi-step RAG is still oftentimes insufficient to accomplish many scenarios of knowledge integration, as we illustrate in what we call *path-finding* multi-hop questions in Figure 1.

In contrast, our brains are capable of solving challenging knowledge integration tasks like these with relative ease. The hippocampal memory indexing theory [75], a well-established theory of human long-term memory, offers one plausible explanation for this remarkable ability. Teyler and Discenna [75] propose that our powerful context-based, continually updating memory relies on interactions between the neocortex, which processes and stores actual memory representations, and the C-shaped hippocampus, which holds the *hippocampal index*, a set of interconnected indices which point to memory units on the neocortex and stores associations between them [19, 76].

In this work, we propose HippoRAG, a RAG framework that serves as a long-term memory for LLMs by mimicking this model of human memory. Our novel design first models the neocortex's ability to process perceptual input by using an LLM to transform a corpus into a schemaless knowledge graph (KG) as our artificial hippocampal index. Given a new query, HippoRAG identifies the key concepts in the query and runs the Personalized PageRank (PPR) algorithm [30] on the KG, using the query concepts as the seeds, to integrate information across passages for retrieval. PPR enables HippoRAG to explore KG paths and identify relevant subgraphs, essentially performing multi-hop reasoning in a single retrieval step.

This capacity for *single-step multi-hop* retrieval yields strong performance improvements of around 3 and 20 points over current RAG methods [10, 35, 53, 70, 71] on two popular multi-hop QA benchmarks, MuSiQue [77] and 2WikiMultiHopQA [33]. Additionally, HippoRAG's online retrieval process is 10 to 30 times cheaper and 6 to 13 times faster than current iterative retrieval methods like IRCoT [78], while still achieving comparable performance. Furthermore, our approach can be combined with IRCoT to provide complementary gains of up to 4% and 20% on the same datasets and even obtain improvements on HotpotQA, a less challenging multi-hop QA dataset. Finally, we

provide a case study illustrating the limitations of current methods as well as our method's potential on the previously discussed *path-finding* multi-hop QA setting.

## 2 HippoRAG

In this section, we first give a brief overview of the hippocampal memory indexing theory, followed by how HippoRAG's indexing and retrieval design was inspired by this theory, and finally offer a more detailed account of our methodology.

### 2.1 The Hippocampal Memory Indexing Theory

The hippocampal memory indexing theory [75] is a well-established theory that provides a functional description of the components and circuitry involved in human long-term memory. In this theory, Teyler and Discenna [75] propose that human long-term memory is composed of three components that work together to accomplish two main objectives: *pattern separation*, which ensures that the representations of distinct perceptual experiences are unique, and *pattern completion*, which enables the retrieval of complete memories from partial stimuli [19, 76].

The theory suggests that pattern separation is primarily accomplished in the memory encoding process, which starts with the **neocortex** receiving and processing perceptual stimuli into more easily manipulatable, likely higher-level, features, which are then routed through the **parahippocampal regions** (PHR) to be indexed by the hippocampus. When they reach the **hippocampus**, salient signals are included in the hippocampal index and associated with each other.

After the memory encoding process is completed, pattern completion drives the memory retrieval process whenever the hippocampus receives partial perceptual signals from the PHR pipeline. The hippocampus then leverages its context-dependent memory system, thought to be implemented through a densely connected network of neurons in the CA3 sub-region [76], to identify complete and relevant memories within the hippocampal index and route them back through the PHR for simulation in the neocortex. Thus, this complex process allows for new information to be integrated by changing only the hippocampal index instead of updating neocortical representations.

### 2.2 Overview

Our proposed approach, HippoRAG, is closely inspired by the process described above. As shown in Figure 2, each component of our method corresponds to one of the three components of human long-term memory. A detailed example of the HippoRAG process can be found in Appendix A.

**Offline Indexing.** Our offline indexing phase, analogous to memory encoding, starts by leveraging a strong instruction-tuned **LLM**, our artificial neocortex, to extract knowledge graph (KG) triples. The KG is schemaless and this process is known as open information extraction (OpenIE) [3, 5, 60, 98]. This process extracts salient signals from passages in a retrieval corpus as discrete noun phrases rather than dense vector representations, allowing for more fine-grained pattern separation. It is therefore natural to define our artificial hippocampal index as this open **KG**, which is built on the whole retrieval corpus passage-by-passage. Finally, to connect both components as is done by the parahippocampal regions, we use off-the-shelf dense encoders fine-tuned for retrieval (**retrieval encoders**). These retrieval encoders provide additional edges between similar but not identical noun phrases within this KG to aid in downstream pattern completion.

**Online Retrieval.** These same three components are then leveraged to perform online retrieval by mirroring the human brain's memory retrieval process. Just as the hippocampus receives input processed through the neocortex and PHR, our LLM-based neocortex extracts a set of salient named entities from a query which we call *query named entities*. These named entities are then linked to nodes in our KG based on the similarity determined by retrieval encoders; we refer to these selected nodes as *query nodes*. Once the query nodes are chosen, they become the partial cues from which our synthetic hippocampus performs pattern completion. In the hippocampus, neural pathways between elements of the hippocampal index enable relevant neighborhoods to become activated and recalled upstream. To imitate this efficient graph search process, we leverage the Personalized PageRank (PPR) algorithm [30], a version of PageRank that distributes probability across a graph only through a set of user-defined source nodes. This constraint allows us to bias the PPR output only towards the

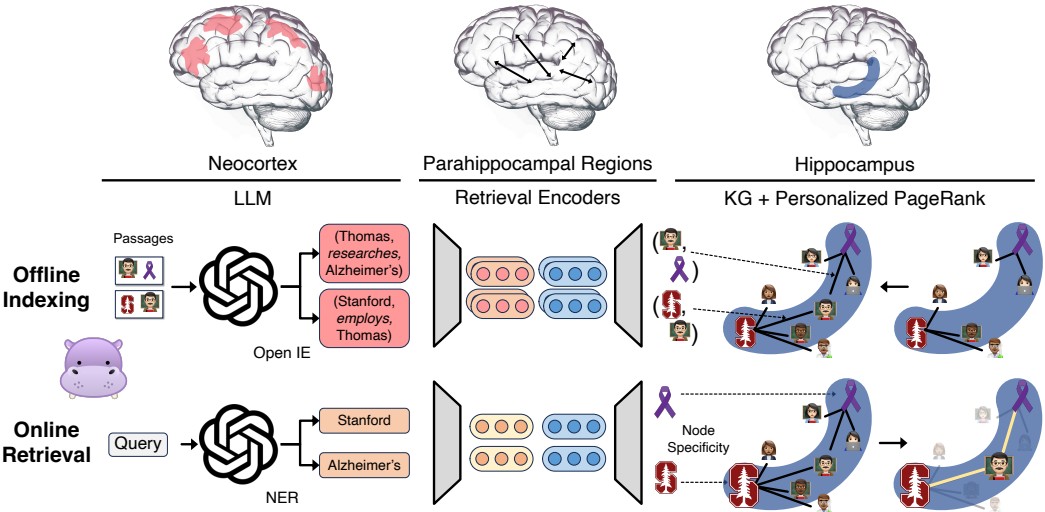

Figure 2: **Detailed HippoRAG Methodology.** We model the three components of human long-term memory to mimic its pattern separation and completion functions. For offline indexing **(Middle)**, we use an LLM to process passages into open KG triples, which are then added to our artificial hippocampal index, while our synthetic parahippocampal regions (PHR) detect synonymy. In the example above, triples involving Professor Thomas are extracted and integrated into the KG. For online retrieval **(Bottom)**, our LLM neocortex extracts named entities from a query while our parahippocampal retrieval encoders link them to our hippocampal index. We then leverage the Personalized PageRank algorithm to enable context-based retrieval and extract Professor Thomas.[4]

set of query nodes, just as the hippocampus extracts associated signals from specific partial cues.[2] Finally, as is done when the hippocampal signal is sent upstream, we aggregate the output PPR node probability over the previously indexed passages and use that to rank them for retrieval.

## 2.3 Detailed Methodology

**Offline Indexing.** Our indexing process involves processing a set of passages $P$ using an instruction-tuned LLM $L$ and a retrieval encoder $M$. As seen in Figure 2 we first use $L$ to extract a set of noun phrase nodes $N$ and relation edges $E$ from each passage in $P$ via OpenIE. This process is done via 1-shot prompting of the LLM with the prompts shown in Appendix I. Specifically, we first extract a set of named entities from each passage. We then add the named entities to the OpenIE prompt to extract the final triples, which also contain concepts (noun phrases) beyond named entities. We find that this two-step prompt configuration leads to an appropriate balance between generality and bias towards named entities. Finally, we use $M$ to add the extra set of *synonymy* relations $E'$ discussed above when the cosine similarity between two entity representations in $N$ is above a threshold $\tau$. As stated above, this introduces more edges to our hippocampal index and allows for more effective pattern completion. This indexing process defines a $|N| \times |P|$ matrix $\mathbf{P}$, which contains the number of times each noun phrase in the KG appears in each original passage.

**Online Retrieval.** During the retrieval process, we prompt $L$ using a 1-shot prompt to extract a set of named entities from a query $q$, our previously defined query named entities $C_q = \{c_1, ..., c_n\}$ (*Stanford* and *Alzheimer's* in our Figure 2 example). These named entities $C_q$ from the query are then encoded by the same retrieval encoder $M$. Then, the previously defined query nodes are chosen as the set of nodes in $N$ with the highest cosine similarity to the query named entities $C_q$. More formally, query nodes are defined as $R_q = \{r_1, ..., r_n\}$ such that $r_i = e_k$ where $k = \arg\max_j cosine\_similarity(M(c_i), M(e_j))$, represented as the *Stanford* logo and the *Alzheimer's* purple ribbon symbol in Figure 2.

---

[2]Intriguingly, some work in cognitive science has also found a correlation between human word recall and the output of the PageRank algorithm [25].

[4]Many details around the hippocampal memory indexing theory are omitted from this study for simplicity. We encourage interested reader to follow the references in §2.1 for more.

After the query nodes $R_q$ are found, we run the PPR algorithm over the hippocampal index, i.e., a KG with $|N|$ nodes and $|E| + |E'|$ edges (triple-based and synonymy-based), using a personalized probability distribution $\vec{n}$ defined over $N$, in which each query node has equal probability and all other nodes have a probability of zero. This allows probability mass to be distributed to nodes that are primarily in the (joint) neighborhood of the query nodes, such as *Professor Thomas*, and contribute to eventual retrieval. After running the PPR algorithm, we obtain an updated probability distribution $\vec{n'}$ over $N$. Finally, in order to obtain passage scores, we multiply $\vec{n'}$ with the previously defined $\mathbf{P}$ matrix to obtain $\vec{p}$, a ranking score for each passage, which we use for retrieval.

**Node Specificity.** We introduce node specificity as a neurobiologically plausible way to further improve retrieval. It is well known that global signals for word importance, like inverse document frequency (IDF), can improve information retrieval. However, in order for our brain to leverage IDF for retrieval, the number of total "passages" encoded would need to be aggregated with all node activations before memory retrieval is complete. While simple for normal computers, this process would require activating connections between an aggregator neuron and all nodes in the hippocampal index every time retrieval occurs, likely introducing prohibitive computational overhead. Given these constraints, we propose *node specificity* as an alternative IDF signal which requires only local signals and is thus more neurobiologically plausible. We define the node specificity of node $i$ as $s_i = |P_i|^{-1}$, where $P_i$ is the set of passages in $P$ from which node $i$ was extracted, information that is already available at each node. Node specificity is used in retrieval by multiplying each query node probability $\vec{n}$ with $s_i$ before PPR; this allows us to modulate each of their neighborhood's probability as well as their own. We illustrate node specificity in Figure 2 through relative symbol size: the *Stanford* logo grows larger than the *Alzheimer's* symbol since it appears in fewer documents.

# 3 Experimental Setup

## 3.1 Datasets

We evaluate our method's retrieval capabilities primarily on two challenging multi-hop QA benchmarks, **MuSiQue** (answerable) [77] and **2WikiMultiHopQA** [33]. For completeness, we also include the **HotpotQA** [89] dataset even though it has been found to be a much weaker test for multi-hop reasoning due to many spurious signals [77], as we also show in Appendix B. To limit the experimental cost, we extract 1,000 questions from each validation set as done in previous work [63, 78]. In order to create a more realistic retrieval setting, we follow IRCoT [78] and collect all candidate passages (including supporting and distractor passages) from our selected questions and form a retrieval corpus for each dataset. The details of these datasets are shown in Table 1.

Table 1: Retrieval corpora and extracted KG statistics for each of our 1,000 question dev sets.

| | MuSiQue | 2Wiki | HotpotQA |
|---|---|---|---|
| # of Passages ($P$) | 11,656 | 6,119 | 9,221 |
| # of Unique Nodes ($N$) | 91,729 | 42,694 | 82,157 |
| # of Unique Edges ($E$) | 21,714 | 7,867 | 17,523 |
| # of Unique Triples | 107,448 | 50,671 | 98,709 |
| # of Contriever Synonym Edges ($E'$) | 145,990 | 146,020 | 159,112 |
| # of ColBERTv2 Synonym Edges ($E'$) | 191,636 | 82,526 | 171,856 |

## 3.2 Baselines

We compare against several strong and widely used retrieval methods: **BM25** [69], **Contriever** [35], **GTR** [53] and **ColBERTv2** [70]. Additionally, we compare against two recent LLM-augmented baselines: **Propositionizer** [10], which rewrites passages into propositions, and **RAPTOR** [71], which constructs summary nodes to ease retrieval from long documents. In addition to the single-step retrieval methods above, we also include the multi-step retrieval method **IRCoT** [78] as a baseline.

## 3.3 Metrics

We report retrieval and QA performance on the datasets above using recall@2 and recall@5 (R@2 and R@5 below) for retrieval and exact match (EM) and F1 scores for QA performance.

Table 2: **Single-step retrieval performance.** HippoRAG outperforms all baselines on MuSiQue and 2WikiMultiHopQA and achieves comparable performance on the less challenging HotpotQA dataset.

| | MuSiQue | | 2Wiki | | HotpotQA | | Average | |
|---|---|---|---|---|---|---|---|---|
| | R@2 | R@5 | R@2 | R@5 | R@2 | R@5 | R@2 | R@5 |
| BM25 [69] | 32.3 | 41.2 | 51.8 | 61.9 | 55.4 | 72.2 | 46.5 | 58.4 |
| Contriever [35] | 34.8 | 46.6 | 46.6 | 57.5 | 57.2 | 75.5 | 46.2 | 59.9 |
| GTR [53] | 37.4 | 49.1 | 60.2 | 67.9 | 59.4 | 73.3 | 52.3 | 63.4 |
| ColBERTv2 [70] | 37.9 | 49.2 | 59.2 | 68.2 | **64.7** | **79.3** | 53.9 | 65.6 |
| RAPTOR [71] | 35.7 | 45.3 | 46.3 | 53.8 | 58.1 | 71.2 | 46.7 | 56.8 |
| RAPTOR (ColBERTv2) | 36.9 | 46.5 | 57.3 | 64.7 | 63.1 | 75.6 | 52.4 | 62.3 |
| Proposition [10] | 37.6 | 49.3 | 56.4 | 63.1 | 58.7 | 71.1 | 50.9 | 61.2 |
| Proposition (ColBERTv2) | 37.8 | 50.1 | 55.9 | 64.9 | 63.9 | 78.1 | 52.5 | 64.4 |
| HippoRAG (Contriever) | **41.0** | **52.1** | **71.5** | **89.5** | 59.0 | 76.2 | 57.2 | 72.6 |
| HippoRAG (ColBERTv2) | 40.9 | 51.9 | 70.7 | 89.1 | 60.5 | 77.7 | **57.4** | **72.9** |

Table 3: **Multi-step retrieval performance.** Combining HippoRAG with standard multi-step retrieval methods like IRCoT results in strong complementary improvements on all three datasets.

| | MuSiQue | | 2Wiki | | HotpotQA | | Average | |
|---|---|---|---|---|---|---|---|---|
| | R@2 | R@5 | R@2 | R@5 | R@2 | R@5 | R@2 | R@5 |
| IRCoT + BM25 (Default) | 34.2 | 44.7 | 61.2 | 75.6 | 65.6 | 79.0 | 53.7 | 66.4 |
| IRCoT + Contriever | 39.1 | 52.2 | 51.6 | 63.8 | 65.9 | 81.6 | 52.2 | 65.9 |
| IRCoT + ColBERTv2 | 41.7 | 53.7 | 64.1 | 74.4 | **67.9** | 82.0 | 57.9 | 70.0 |
| IRCoT + HippoRAG (Contriever) | 43.9 | 56.6 | 75.3 | 93.4 | 65.8 | 82.3 | 61.7 | 77.4 |
| IRCoT + HippoRAG (ColBERTv2) | **45.3** | **57.6** | **75.8** | **93.9** | 67.0 | **83.0** | **62.7** | **78.2** |

## 3.4 Implementation Details

By default, we use `GPT-3.5-turbo-1106` [55] with temperature of $0$ as our LLM $L$ and Contriever [35] or ColBERTv2 [70] as our retriever $M$. We use $100$ examples from MuSiQue's training data to tune HippoRAG's two hyperparameters: the synonymy threshold $\tau$ at $0.8$ and the PPR damping factor at $0.5$, which determines the probability that PPR will restart a random walk from the query nodes instead of continuing to explore the graph. Generally, we find that HippoRAG's performance is rather robust to its hyperparameters. More implementation details can be found in Appendix H.

## 4 Results

We present our retrieval and QA experimental results below. Given that our method indirectly affects QA performance, we report QA results on our best-performing retrieval backbone ColBERTv2 [70]. However, we report retrieval results for several strong single-step and multi-step retrieval techniques.

**Single-Step Retrieval Results.** As seen in Table 2, HippoRAG outperforms all other methods, including recent LLM-augmented baselines such as Propositionizer and RAPTOR, on our main datasets, MuSiQue and 2WikiMultiHopQA, while achieving competitive performance on HotpotQA. We notice an impressive improvement of $11$ and $20\%$ for R@2 and R@5 on 2WikiMultiHopQA and around $3\%$ on MuSiQue. This difference can be partially explained by 2WikiMultiHopQA's entity-centric design, which is particularly well-suited for HippoRAG. Our lower performance on HotpotQA is mainly due to its lower knowledge integration requirements, as explained in Appendix B, as well as a due to a concept-context tradeoff which we alleviate with an ensembling technique described in Appendix F.2.

**Multi-Step Retrieval Results.** For multi-step or iterative retrieval, our experiments in Table 3 demonstrate that IRCoT [78] and HippoRAG are complementary. Using HippoRAG as the retriever for IRCoT continues to bring R@5 improvements of around $4\%$ for MuSiQue, $18\%$ for 2WikiMultiHopQA and an additional $1\%$ on HotpotQA.

Table 4: **QA performance.** HippoRAG's QA improvements correlate with its retrieval improvements on single-step (rows 1-3) and multi-step retrieval (rows 4-5).

| | MuSiQue | | 2Wiki | | HotpotQA | | Average | |
|---|---|---|---|---|---|---|---|---|
| Retriever | EM | F1 | EM | F1 | EM | F1 | EM | F1 |
| None | 12.5 | 24.1 | 31.0 | 39.6 | 30.4 | 42.8 | 24.6 | 35.5 |
| ColBERTv2 | 15.5 | 26.4 | 33.4 | 43.3 | 43.4 | 57.7 | 30.8 | 42.5 |
| HippoRAG (ColBERTv2) | 19.2 | 29.8 | 46.6 | 59.5 | 41.8 | 55.0 | 35.9 | 48.1 |
| IRCoT (ColBERTv2) | 19.1 | 30.5 | 35.4 | 45.1 | 45.5 | 58.4 | 33.3 | 44.7 |
| IRCoT + HippoRAG (ColBERTv2) | **21.9** | **33.3** | **47.7** | **62.7** | **45.7** | **59.2** | **38.4** | **51.7** |

**Question Answering Results.** We report QA results for HippoRAG, the strongest retrieval baselines, ColBERTv2 and IRCoT, as well as IRCoT using HippoRAG as a retriever in Table 4. As expected, improved retrieval performance in both single and multi-step settings leads to strong overall improvements of up to 3%, 17% and 1% F1 scores on MuSiQue, 2WikiMultiHopQA and HotpotQA respectively using the same QA reader. Notably, single-step HippoRAG is on par or outperforms IRCoT while being 10-30 times cheaper and 6-13 times faster during online retrieval (Appendix G).

## 5 Discussions

### 5.1 What Makes HippoRAG Work?

**OpenIE Alternatives.** To determine if using a closed model like GPT-3.5 is essential to retain our performance improvements, we replace it with an end-to-end OpenIE model REBEL [34] as well as the 8B and 70B instruction-tuned versions of Llama-3.1, a class of strong open-weight LLMs [1]. As shown in Table 5 row 2, building our KG using REBEL results in large performance drops, underscoring the importance of LLM flexibility. Specifically, GPT-3.5 produces twice as many triples as REBEL, indicating its bias against producing triples with general concepts and leaving many useful associations behind.

In terms of open-weight LLMs, Table 5 (rows 3-4) shows that the performance of Llama-3.1-8B is competitive with GPT-3.5 in all datasets except for 2Wiki, where performance drops substantially. Nevertheless, the stronger 70B counterpart outperforms GPT-3.5 in two out of three datasets and is still competitive in 2Wiki. The strong performance of Llama-3.1-70B and the comparable performance of even the 8B model is encouraging since it offers a cheaper alternative for indexing over large corpora. The graph statistics for these OpenIE alternatives can be found in Appendix C.

To understand the relationship between OpenIE and retrieval performance more deeply, we extract 239 gold triples from 20 examples from the MuSiQue training set. We then perform a small-scale intrinsic evaluation using the CaRB [6] framework for OpenIE. We find that both Llama-3.1-Instruct

Table 5: **Dissecting HippoRAG.** To understand what makes it work well, we replace its OpenIE module and PPR with plausible alternatives and ablate node specificity and synonymy-based edges.

| | | MuSiQue | | 2Wiki | | HotpotQA | | Average | |
|---|---|---|---|---|---|---|---|---|---|
| | | R@2 | R@5 | R@2 | R@5 | R@2 | R@5 | R@2 | R@5 |
| HippoRAG | | 40.9 | 51.9 | **70.7** | **89.1** | 60.5 | 77.7 | **57.4** | **72.9** |
| OpenIE Alternatives | REBEL [34] | 31.7 | 39.6 | 63.1 | 76.5 | 43.9 | 59.2 | 46.2 | 58.4 |
| | Llama-3.1-8B-Instruct [1] | 40.8 | 51.9 | 62.5 | 77.5 | 59.9 | 75.1 | 54.4 | 67.8 |
| | Llama-3.1-70B-Instruct [1] | **41.8** | **53.7** | 68.8 | 85.3 | **60.8** | **78.6** | 57.1 | 72.5 |
| PPR Alternatives | $R_q$ Nodes Only | 37.1 | 41.0 | 59.1 | 61.4 | 55.9 | 66.2 | 50.7 | 56.2 |
| | $R_q$ Nodes & Neighbors | 25.4 | 38.5 | 53.4 | 74.7 | 47.8 | 64.5 | 42.2 | 59.2 |
| Ablations | w/o Node Specificity | 37.6 | 50.2 | 70.1 | 88.8 | 56.3 | 73.7 | 54.7 | 70.9 |
| | w/o Synonymy Edges | 40.2 | 50.2 | 69.2 | 85.6 | 59.1 | 75.7 | 56.2 | 70.5 |

models underperform GPT-3.5 slightly on this intrinsic evaluation but all LLMs vastly outperform REBEL. More details about this evaluation experiments can be found in Appendix D.

**PPR Alternatives.** As shown in Table 5 (rows 5-6), to examine how much of our results are due to the strength of PPR, we replace the PPR output with the query node probability $\vec{n}$ multiplied by node specificity values (row 5) and a version of this that also distributes a small amount of probability to the direct neighbors of each query node (row 6). First, we find that PPR is a much more effective method for including associations for retrieval on all three datasets compared to both simple baselines. It is interesting to note that adding the neighborhood of $R_q$ nodes without PPR leads to worse performance than only using the query nodes themselves.

**Ablations.** As seen in Table 5 (rows 7-8), node specificity obtains considerable improvements on MuSiQue and HotpotQA and yields almost no change in 2WikiMultiHopQA. This is likely because 2WikiMultiHopQA relies on named entities with little differences in terms of term weighting. In contrast, synonymy edges have the largest effect on 2WikiMultiHopQA, suggesting that noisy entity standardization is useful when most relevant concepts are named entities, and improvements to synonymy detection could lead to stronger performance in other datasets.

### 5.2 HippoRAG's Advantage: Single-Step Multi-Hop Retrieval

A major advantage of HippoRAG over conventional RAG methods in multi-hop QA is its ability to *perform multi-hop retrieval in a single step*. We demonstrate this by measuring the percentage of queries where *all* the supporting passages are retrieved successfully, a feat that can only be accomplished through successful multi-hop reasoning. Table 6 below shows that the gap between our method and ColBERTv2, using the top-5 passages, increases even more from 3% to 6% on MuSiQue and from 20% to 38% on 2WikiMultiHopQA, suggesting that large improvements come from obtaining all supporting documents rather than achieving partially retrieval on more questions.

Table 6: **All-Recall metric.** We measure the percentage of queries for which all supporting passages are successfully retrieved (all-recall, denoted as AR@2 or AR@5) and find even larger performance improvements for HippoRAG.

| | MuSiQue | | 2Wiki | | HotpotQA | | Average | |
|---|---|---|---|---|---|---|---|---|
| | AR@2 | AR@5 | AR@2 | AR@5 | AR@2 | AR@5 | AR@2 | AR@5 |
| ColBERTv2 [70] | 6.8 | 16.1 | 25.1 | 37.1 | 33.3 | 59.0 | 21.7 | 37.4 |
| HippoRAG | 10.2 | 22.4 | 45.4 | 75.7 | 33.8 | 57.9 | 29.8 | 52.0 |

We further illustrate HippoRAG's unique *single-step multi-hop retrieval* ability through the first example in Table 7. In this example, even though *Alhandra* was not mentioned in *Vila de Xira's* passage, HippoRAG can directly leverage Vila de Xira's connection to Alhandra as his place of birth to determine its importance, something that standard RAG methods would be unable to do directly. Additionally, even though IRCoT can also solve this multi-hop retrieval problem, as shown in Appendix G, it is 10-30 times more expensive and 6-13 times slower than ours in terms of online retrieval, arguably the most important factor when it comes to serving end users.

Table 7: **Multi-hop question types.** We show example results for different approaches on path-finding vs. path-following multi-hop questions.

| | Question | HippoRAG | ColBERTv2 | IRCoT |
|---|---|---|---|---|
| Path-Following | In which district was **Alhandra** born? | **1. Alhandra** **2. Vila de Xira** 3. Portugal | **1. Alhandra** 2. Dimuthu Abayakoon 3. Ja'ar | **1. Alhandra** **2. Vila de Xira** 3. Póvoa de Santa Iria |
| Path-Finding | Which **Stanford** professor works on the neuroscience of **Alzheimer's**? | **1. Thomas Südhof** **2. Karl Deisseroth** **3. Robert Sapolsky** | 1. Brian Knutson 2. Eric Knudsen 3. Lisa Giocomo | 1. Brian Knutson 2. Eric Knudsen 3. Lisa Giocomo |

### 5.3 HippoRAG's Potential: Path-Finding Multi-Hop Retrieval

The second example in Table 7, also present in Figure 1, shows a type of questions that is trivial for informed humans but out of reach for current retrievers without further training. This type of questions, which we call *path-finding* multi-hop questions, requires identifying one path between a set of entities when many paths exist to explore instead of *following* a specific path, as in standard multi-hop questions.[5]

More specifically, a simple iterative process can retrieve the appropriate passages for the first question by following the one path set by *Alhandra's* one place of birth, as seen by IRCoT's perfect performance. However, an iterative process would struggle to answer the second question given the many possible paths to explore—either through professors at *Stanford University* or professors working on the neuroscience of *Alzheimer's*. It is only by associating disparate information about Thomas Südhof that someone who knows about this professor would be able to answer this question easily. As seen in Table 7, both ColBERTv2 and IRCoT fail to extract the necessary passages since they cannot access these associations. On the other hand, HippoRAG leverages its web of associations in its hippocampal index and graph search algorithm to determine that Professor Thomas is relevant to this query and retrieves his passages appropriately. More examples of these path-finding multi-hop questions can be found in our case study in Appendix E.

## 6 Related Work

### 6.1 LLM Long-Term Memory

**Parametric Long-Term Memory.** It is well-accepted, even among skeptical researchers, that the parameters of modern LLMs encode a remarkable amount of world knowledge [2, 12, 23, 28, 31, 39, 62, 79], which can be leveraged by an LLM in flexible and robust ways [81, 83, 93]. Nevertheless, our ability to update this vast knowledge store, an essential part of any long-term memory system, is still surprisingly limited. Although many techniques to update LLMs exist, such as standard fine-tuning, model editing [15, 49, 50, 51, 52, 95] and even external parametric memory modules inspired by human memory [58, 82, 32], no methodology has yet to emerge as a robust solution for continual learning in LLMs [26, 46, 97].

**RAG as Long-Term Memory.** On the other hand, using RAG methods as a long-term memory system offers a simple way to update knowledge over time [36, 42, 66, 73]. More sophisticated RAG methods, which perform multiple steps of retrieval and generation from an LLM, are even able to integrate information across new or updated knowledge elements[38, 64, 72, 78, 88, 90, 92], another crucial aspect of long-term memory systems. As discussed above, however, this type of online information integration is unable to solve the more complex knowledge integration tasks that we illustrate with our *path-finding* multi-hop QA examples.

Some other methods, such as RAPTOR [71], MemWalker [9] and GraphRAG [18], integrate information during the offline indexing phase similarly to HippoRAG and might be able to handle these more complex tasks. However, these methods integrate information by summarizing knowledge elements, which means that the summarization process must be repeated any time new data is added. In contrast, HippoRAG can continuously integrate new knowledge by simply adding edges to its KG.

**Long Context as Long-Term Memory.** Context lengths for both open and closed source LLMs have increased dramatically in the past year [11, 17, 22, 61, 68]. This scaling trend seems to indicate that future LLMs could perform long-term memory storage within massive context windows. However, the viability of this future remains largely uncertain given the many engineering hurdles involved and the apparent limitations of long-context LLMs, even within current context lengths [41, 45, 96, 21].

### 6.2 Multi-Hop QA & Graphs

Many previous works have also tackled multi-hop QA using graph structures. These efforts can be broadly divided in two major categories: 1) graph-augmented reading comprehension, where a

---

[5]Path-finding questions require knowledge integration when search entities like *Stanford* and *Alzheimer's* do not happen to appear together in a passage, a condition which is often satisfied for new information.

graph is extracted from retrieved documents and used to improve a model's reasoning process and 2) graph-augmented retrieval, where models find relevant documents by traversing a graph structure.

**Graph-Augmented Reading Comprehension.** Earlier works in this category are mainly supervised methods which mix signal from a hyperlink or co-occurrence graph with a language model through a graph neural network (GNN) [20, 67, 65]. More recent works use LLMs and introduce knowledge graph triples directly into the LLM prompt [57, 43, 47]. Although these works share HippoRAG's use of graphs for multi-hop QA, their generation-based improvements are fully complementary to HippoRAG's, which are solely based on improved retrieval.

**Graph-Augmented Retrieval.** In this second category, previous work trains a re-ranking module which can traverse a graph made using Wikipedia hyperlinks [16, 100, 54, 14, 4, 44]. HippoRAG, in contrast, builds a KG from scratch using LLMs and performs multi-hop retrieval without any supervision, making it much more adaptable.

### 6.3 LLMs & KGs

Combining the strengths of language models and knowledge graphs has been an active research direction for many years, both for augmenting LLMs with a KG in different ways [48, 80, 84] or augmenting KGs by either distilling knowledge from an LLM's parametric knowledge [7, 85] or using them to parse text directly [8, 29, 94]. In an exceptionally comprehensive survey, Pan et al. [56] present a roadmap for this research direction and highlight the importance of work which *synergizes* these two important technologies [37, 74, 27, 91, 99]. Like these works, HippoRAG shows the potential for synergy between these two technologies, combining the knowledge graph construction abilities of LLMs with the retrieval advantages of structured knowledge for more effective RAG.

## 7 Conclusions & Limitations

Our proposed neurobiologically principled methodology, although simple, already shows promise for overcoming the inherent limitations of standard RAG systems while retaining their advantages over parametric memory. HippoRAG's knowledge integration capabilities, demonstrated by its strong results on *path-following* multi-hop QA and promise on *path-finding* multi-hop QA, as well as its dramatic efficiency improvements and continuously updating nature, makes it a powerful middle-ground framework between standard RAG methods and parametric memory and offers a compelling solution for long-term memory in LLMs.

Nevertheless, several limitations can be addressed in future work to enable HippoRAG to achieve this goal better. First, we note that all components of HippoRAG are currently used off-the-shelf without any extra training. There is therefore much room to improve our method's practical viability by performing specific component fine-tuning. This is evident in the error analysis discussed in Appendix F, which shows most errors made by our system are due to NER and OpenIE and thus could benefit from direct fine-tuning. Given that the rest of the errors are graph search errors, also in Appendix F, we note that several avenues for improvements over simple PPR exist, such as allowing relations to guide graph traversal directly. Additionally, as shown in Appendix F.4, more work must be done to improve the consistency of OpenIE in longer compared to shorter documents. Finally, and perhaps most importantly, HippoRAG's scalability still calls for further validation. Although we show that Llama-3.1 could obtain similar performance to closed-source models and thus reduce costs considerably, we are yet to empirically prove the efficiency and efficacy of our synthetic hippocampal index as its size grows way beyond current benchmarks.

## Acknowledgments

The authors would like to thank colleagues from the OSU NLP group and Percy Liang for their thoughtful comments. This research was supported in part by NSF OAC 2112606, NIH R01LM014199, ARL W911NF2220144, and Cisco. The views and conclusions contained herein are those of the authors and should not be interpreted as representing the official policies, either expressed or implied, of the U.S. government. The U.S. Government is authorized to reproduce and distribute reprints for Government purposes notwithstanding any copyright notice herein.

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

# Appendices

Within this supplementary material, we elaborate on the following aspects:

- Appendix A: HippoRAG Pipeline Example
- Appendix B: Dataset Comparison
- Appendix C: Ablation Statistics
- Appendix D: Intrinsic OpenIE Evaluation
- Appendix E: Path-Finding Multi-Hop Case Study
- Appendix F: Error Analysis
- Appendix G: Cost and Efficiency Comparison
- Appendix H: Implementation Details & Compute Requirements
- Appendix I: LLM Prompts

## Question & Answer

**Question**    In which district was Alhandra born?
**Answer**    Lisbon

## Supporting Passages

**1. Alhandra (footballer)**

Luís Miguel Assunção Joaquim (born 5 March 1979 in Vila Franca de Xira, Lisbon), known as Alhandra, is a Portuguese retired footballer who played mainly as a left back – he could also appear as a midfielder.

**2. Vila Franca de Xira**

Vila Franca de Xira is a municipality in the Lisbon District in Portugal. The population in 2011 was 136,886, in an area of 318.19 km². Situated on both banks of the Tagus River, 32 km north-east of the Portuguese capital Lisbon, settlement in the area dates back to neolithic times, as evidenced by findings in the Cave of Pedra Furada. Vila Franca de Xira is said to have been founded by French followers of Portugal's first king, Afonso Henriques, around 1200.

## Distractor Passages (Excerpts)

**1. Chirakkalkulam**
Chirakkalkulam is a small residential area near Kannur town of Kannur District, Kerala state, South India. Chirakkalkulam is located between Thayatheru and Kannur City. Chirakkalkulam's significance arises from the birth of the historic Arakkal Kingdom.

**2. Frank T. and Polly Lewis House**
The Frank T. and Polly Lewis House is located in Lodi, Wisconsin, United States. It was added to the National Register of Historic Places in 2009. The house is located within the Portage Street Historic District.

**3. Birth certificate**
In the U.S., the issuance of birth certificates is a function of the Vital Records Office of the states, capital district, territories and former territories …

Figure 3: **HippoRAG Pipeline Example (Question and Annotations). (Top)** We provide an example question and its answer. **(Middle & Bottom)** The supporting and distractor passages for this question. Two supporting passages are needed to solve this question. The excerpts of the distractor passages are related to the "district" mentioned in the question.

## A HippoRAG Pipeline Example

To better demonstrate how our HippoRAG pipeline works, we use the *path-following* example from the MuSiQue dataset shown in Table 7. We use HippoRAG's indexing and retrieval processes to follow this question and a subset of the associated corpus. The question, its answer, and its supporting and distractor passages are as shown in Figure 3. The indexing stage is shown in Figure 4, showing both the OpenIE procedure as well as the relevant subgraph of our KG. Finally, we illustrate the retrieval stage in Figure 5, including query NER, query node retrieval, how the PPR algorithm changes node probabilities, and how the top retrieval results are calculated.

## Indexing: Passage NER and OpenIE for Supporting Passages

**1. Alhandra (footballer)**
NER:
["5 March 1979", "Alhandra", "Lisbon", "Luís Miguel Assunção Joaquim", "Portuguese", "Vila Franca de Xira"]

OpenIE:
[("Alhandra", "is a", "footballer"),
("Alhandra", "born in", "Vila Franca de Xira"),
("Alhandra", "born in", "Lisbon"),
("Alhandra", "born on", "5 March 1979"),
("Alhandra", "is", "Portuguese"),
("Luís Miguel Assunção Joaquim", "is also known as", "Alhandra")]

**2. Vila Franca de Xira**
NER:
["2011", "Afonso Henriques", "Cave of Pedra Furada", "French", "Lisbon", "Lisbon District", "Portugal", "Tagus River", "Vila Franca de Xira"]

OpenIE:
[("Vila Franca de Xira", "is a municipality in", "Lisbon District"),
("Vila Franca de Xira", "located in", "Portugal"),
("Vila Franca de Xira", "situated on", "Tagus River"),
("Vila Franca de Xira", "is", "founded by French followers of Afonso Henriques"),
("Tagus River", "located near", "Lisbon"),
("Cave of Pedra Furada", "evidenced settlement in", "neolithic times"),
("Afonso Henriques", "was Portugal's first king in", "1200"),
("Vila Franca de Xira", "had population of", "136,886 in 2011"),
("Vila Franca de Xira", "has area of", "318.19 km²")]

## Indexing: Subgraph Related to the Question

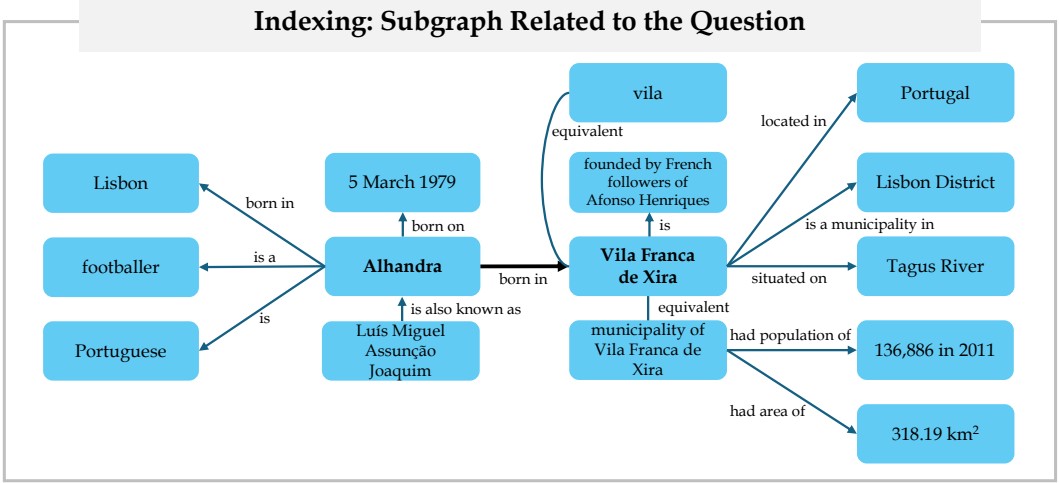

Figure 4: **HippoRAG Pipeline Example (Indexing).** NER and OpenIE are sequentially conducted on each passage of the corpus. Thus, an open knowledge graph is formed for the entire corpus. We only show the relevant subgraph from the KG.

## Retrieval: Query NER & Node Retrieval

**Question**       In which district was Alhandra born?
**NER**            ["Alhandra"]
**Node Retrieval** {"Alhandra": "Alhandra"}

## Retrieval: PPR

**Node Probabilities Changes by PPR**

| | | | |
|---|---|---|---|
| Alhandra | 1.000 ⇒ **0.533** | 5 March 1979 | 0.000 ⇒ 0.045 |
| Vila Franca de Xira | 0.000 ⇒ **0.054** | Luís Miguel Assunção Joaquim | 0.000 ⇒ 0.044 |
| Lisbon | 0.000 ⇒ 0.049 | Portugal | 0.000 ⇒ 0.009 |
| footballer | 0.000 ⇒ 0.047 | Tagus River | 0.000 ⇒ 0.007 |
| Portuguese | 0.000 ⇒ 0.046 | José Pinto Coelho | 0.000 ⇒ 0.004 |
| … | | | |

## Retrieval: Top Results

*Top-ranked nodes from PPR are highlighted.

**1. Alhandra (footballer)**
Luís Miguel Assunção Joaquim (born 5 March 1979 in Vila Franca de Xira, Lisbon), known as Alhandra, is a Portuguese retired footballer who played mainly as a left back – he could also appear as a midfielder.

**2. Vila Franca de Xira**
Vila Franca de Xira is a municipality in the Lisbon District in Portugal. The population in 2011 was 136,886, in an area of 318.19 km². Situated on both banks of the Tagus River, 32 km north-east of the Portuguese capital Lisbon, settlement in the area dates back to neolithic times, as evidenced by findings in the Cave of Pedra Furada. Vila Franca de Xira is said to have been founded by French followers of Portugal's first king, Afonso Henriques, around 1200.

**3. Portugal**
Portuguese is the official language of Portugal. Portuguese is a Romance language that originated in what is now Galicia and Northern Portugal, originating from Galician-Portuguese, which was the common language of the Galician and Portuguese people until the independence of Portugal. Particularly in the North of Portugal, there are still many similarities between the Galician culture and the Portuguese culture. Galicia is a consultative observer of the Community of Portuguese Language Countries. According to the Ethnologue of Languages, Portuguese and Spanish have a lexical similarity of 89% - educated speakers of each language can communicate easily with one another.

**4. Huguenots**
The first Huguenots to leave France sought freedom from persecution in Switzerland and the Netherlands … A fort, named Fort Coligny, was built to protect them from attack from the Portuguese troops and Brazilian Native Americans. It was an attempt to establish a French colony in South America. The fort was destroyed in 1560 by the Portuguese, who captured part of the Huguenots. The Portuguese threatened the prisoners with death if they did not convert to Catholicism …

**5. East Timor**
Democratic Republic of Timor - Leste República Demokrátika Timór Lorosa'e (Tetum) República Democrática de Timor - Leste (Portuguese) Flag Coat of arms Motto: Unidade, Acção, Progresso (Portuguese) Unidade, Asaun, Progresu (Tetum) (English: ``Unity, Action, Progress '') Anthem: Pátria (Portuguese) (English:`` Fatherland'') Capital and largest city Dili 8 ° 20 ′ S 125 ° 20 ′ E  /  8.34 ° S 125.34 ° E  / - 8.34; 125.34 Coordinates: 8 ° 20 ′ S 125 ° 20 ′ E  /  8.34 ° S 125.34 ° E  / - 8.34; 125.34 …

Figure 5: **HippoRAG Pipeline Example (Retrieval).** For retrieval, the named entities in the query are extracted from the question **(Top)**, after which the query nodes are chosen using a retrieval encoder. In this case, the name of the query named entity, "Alhandra", is equivalent to its KG node. **(Middle)** We then set the personalized probabilities for PPR based on the retrieved query nodes. After PPR, the query node probability is distributed according to the subgraph in Figure 4, leading to some probability mass on the node "Vila France de Xira". **(Bottom)** These node probabilities are then summed over the passages they appear in to obtain the passage-level ranking. The top-ranked nodes after PPR are highlighted in the top-ranked passages.

## B Dataset Comparison

To analyze the differences between the three datasets we use, we pay special attention to the quality of the distractor passages, i.e., whether they can be effectively confounded with the supporting passages. We use Contriever [35] to calculate the match score between questions and candidate passages and show their densities in Figure 6. In an ideal case, the distribution of distractor scores should be close to the mean of the support passage scores. However, it can be seen that the distribution of the distractor scores in HotpotQA is much closer to the lower bound of the support passage scores compared to the other two datasets.

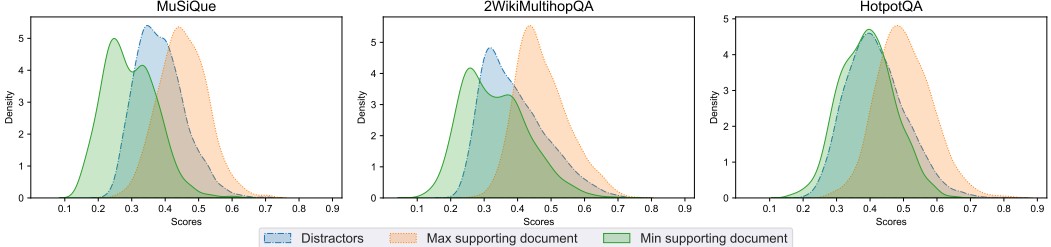

Figure 6: Density of similarity scores of candidate passages (distractors and supporting passages) obtained by Contriever. The similarity score of HotpotQA distractors is not substantially larger than that of the least similar supporting passages, meaning that these distractors are not very effective.

## C Ablation Statistics

We use GPT-3.5 Turbo, REBEL [34] and Llama-3.1 (8B and 70B) [1] for OpenIE ablation experiments. As shown in Table 8, compared to both GPT-3.5 Turbo and both Llama models, REBEL generates around half the number of nodes and edges. This illustrates REBEL's lack of flexibility in open information extraction when compared to using both open and closed-source LLMs. Meanwhile, both Llama-3.1 versions produce a similar amount of OpenIE triples than GPT-3.5 Turbo.

Table 8: Knowledge graph statistics using different OpenIE methods.

| Model | Count | MuSiQue | 2Wiki | HotpotQA |
|---|---|---|---|---|
| GPT-3.5 Turbo (1106) [55] (Default) | # of Unique Nodes ($N$) | 91,729 | 42,694 | 82,157 |
| | # of Unique Edges ($E$) | 21,714 | 7,867 | 17,523 |
| | # of Unique Triples | 107,448 | 50,671 | 98,709 |
| | # of ColBERTv2 Synonym Edges ($E'$) | 191,636 | 82,526 | 171,856 |
| REBEL-large [34] | # of Unique Nodes ($N$) | 36,653 | 22,146 | 30,426 |
| | # of Unique Edges ($E$) | 269 | 211 | 262 |
| | # of Unique Triples | 52,102 | 30,428 | 42,038 |
| | # of ColBERTv2 Synonym Edges ($E'$) | 48,213 | 33,072 | 39,053 |
| Llama-3.1-8B-Instruct [1] | # of Unique Nodes ($N$) | 86,864 | 37,875 | 76,311 |
| | # of Unique Edges ($E$) | 22,807 | 6,729 | 18,109 |
| | # of Unique Triples | 118,430 | 47,420 | 104,981 |
| | # of ColBERTv2 Synonym Edges ($E'$) | 155,889 | 72,963 | 139,181 |
| Llama-3.1-70B-Instruct [1] | # of Unique Nodes ($N$) | 80,634 | 39,845 | 70,304 |
| | # of Unique Edges ($E$) | 22,120 | 6,996 | 16,404 |
| | # of Unique Triples | 120,514 | 55,940 | 105,281 |
| | # of ColBERTv2 Synonym Edges ($E'$) | 140,328 | 69,125 | 119,948 |

## D Intrinsic OpenIE Evaluation

In order to better understand how OpenIE and retrieval interact, we extracted gold triples from 20 documents from the MuSiQue training dataset. In total, we extracted 239 gold triples. From the

results in Table 9, we first note that there is a massive difference between end-to-end information extraction systems like REBEL and LLMs. Additionally, we note that there is some correlation better OpenIE and retrieval performance, given that the 8B Llama-3.1-Instruct version performs worse that its 70B counterpart in both retrieval and intrinsic metrics. More specifically, we see that this larger model only provides intrinsic improvements in the recall metric, which seems specially important in improving retrieval performance. Finally, we note that this evaluation is not perfectly correlated with retrieval performance, since GPT-3.5's intrinsic performance is much stronger than Llama-3.1-70B-Instruct while its retrieval score is only slightly higher.

Table 9: Intrinsic OpenIE evaluation using the CaRB [6] framework on 20 annotated passages.

|  | AUC | Precision | Recall | F1 |
|---|---|---|---|---|
| GPT-3.5 Turbo (1106) [55] (Default) | 46.5 | 68.4 | 55.2 | 61.1 |
| Llama-3.1-8B-Instruct [1] | 40.0 | 66.4 | 48.1 | 55.8 |
| Llama-3.1-70B-Instruct [1] | 42.3 | 66.3 | 50.9 | 57.6 |
| REBEL [34] | 1.0 | 8.0 | 1.8 | 2.9 |

## E Case Study on Path-Finding Multi-Hop QA

As discussed above, path-finding multi-hop questions across passages are exceedingly challenging for single-step and multi-step RAG methods such as ColBERTv2 and IRCoT. These questions require integrating information across multiple passages to find relevant entities among many possible candidates, such as finding all Stanford professors who work on the neuroscience of Alzheimer's.

### E.1 Path-Finding Multi-Hop Question Construction Process

These questions and the curated corpora around them were built through the following procedure. The first two questions follow a slightly separate process as the third one as well as the motivating example in the main paper. For the first two, we first identify a book or movie and then found the book's author or the movie's director. We would then find 1) a trait for either the book/movie and 2) another trait for the author/director. These two traits would then be used to extract distractors from Wikipedia for each question.

For the third question and our motivating example, we first choose a professor or a drug at random as the answer for each question. We then obtain the university the professor works at or the disease the drug treats as well as one other trait for the professor or drug (in these questions research topic and mechanism of action were chosen). In these questions, distractors were extracted from Wikipedia using the University or disease on the one hand and the research topic or mechanism of action on the other. This process, although quite tedious, allowed us to curate these challenging but realistic path-finding multi-hop questions.

### E.2 Qualitative Analysis

In Table 10, we show three more examples from three different domains that illustrate HippoRAG's potential for solving retrieval tasks that require such cross-passage knowledge integration.

In the first question of Table 10, we want to find a book published in **2012** by an English author who won a specific award. In contrast to HippoRAG, ColBERTv2 and IRCoT are unable to identify **Mark Haddon** as such an author. ColBERTv2 focuses on passages related to awards while IRCoT mistakenly decides that Kate Atkinson is the answer to such question since she won the same award for a book published in 1995. For the second question, we wanted to find a war film based on a non-fiction book directed by someone famous for sci-fi and crime movies. HippoRAG is able to find our answer **Black Hawk Down** by **Ridley Scott** within the first four passages, while ColBERTv2 misses the answer completely and retrieves other films and film collections. In this instance, even though IRCoT is able to retrieve Ridley Scott, it does so mainly through parametric knowledge. The chain-of-thought output discusses his and Denis Villeneuve fame as well as their sci-fi and crime experience. Given the three-step iteration restriction used here and the need to explore two directors, the specific war film **Black Hawk Down** was not identified. Although a bit convoluted, people often

ask these first two questions to remember a specific movie or book they watched or heard about from only a handful of disjointed details.

Finally, the third question is more similar to the motivating example in the main paper and shows the importance of this type of question in real-world domains. In this question, we ask for a drug used to treat lymphocytic leukemia through a specific mechanism (cytosolic p53 interaction). While HippoRAG is able to leverage the associations within the supporting passages to identify the **Chlorambucil** passage as the most important, ColBERTv2 and IRCoT are only able to extract passages associated with lymphocytic leukemia. Interestingly enough, IRCoT uses its parametric knowledge to guess that Venetoclax, which also treats leukemia, would do so through the relevant mechanism even though no passage in the curated dataset explicitly stated this.

Table 10: Ranking result examples for different approaches on several path-finding multi-hop questions.

| Question | HippoRAG | ColBERTv2 | IRCoT |
|---|---|---|---|
| Which book was published in **2012** by an **English** author who is a **Whitbread Award** winner? | **1.** Oranges Are Not the Only Fruit **2.** William Trevor Legacies **3. Mark Haddon** | **1.** World Book Club Prize winners **2.** Leon Garfield Awards **3.** Twelve Bar Blues (novel) | **1.** Kate Atkinson **2.** Leon Garfield Awards **3.** Twelve Bar Blues (novel) |
| Which **war film** based on a **non fiction book** was directed by someone famous in the **science fiction** and **crime genres**? | **1.** War Film **2.** Time de Zarn **3.** Outline of Sci-Fi **4. Black Hawk Down** | **1.** Paul Greengrass **2.** List of book-based war films **3.** Korean War Films **4.** All the King's Men Book | **1. Ridley Scott 2.** Peter Hyams **3.** Paul Greengrass **4.** List of book-based war films |
| What drug is used to treat **chronic lymphocytic leukemia** by interacting with **cytosolic p53**? | **1. Chlorambucil 2. Lymphocytic leukemia 3.** Mosquito bite allergy | **1. Lymphocytic leukemia 2.** Obinutuzumab **3.** Venetoclax | **1.** Venetoclax **2. Lymphocytic leukemia 3.** Idelalisib |

# F  Error Analysis

## F.1  Overview

In this section, we provide a detailed error analysis of 100 errors made by HippoRAG on the MuSiQue dataset. As shown in Table 11, these errors can be categorized into three main types: NER, OpenIE and PPR.

The main error type, with nearly half of all error examples, is due to limitations of our NER based design. As further discussed in §F.2, our NER design does not extract enough information from the query for retrieval. For example, in the question "When was one internet browser's version of Windows 8 made accessible?", only the phrase "Windows 8" is extracted, leaving any signal about "browsers" or "accessibility" behind for the subsequent graph search. OpenIE errors, the second most common, are discussed in more detail in §F.3.

We define the third error category as cases where both NER and OpenIE are functioning properly but the PPR algorithm is still unable to identify relevant subgraphs, often due to confounding signals. For instance, consider the query "How many refugees emigrated to the European country where Huguenots felt a kinship for emigration?". Despite the term "Huguenots" being accurately extracted from both the question and the supporting passages, and the PPR algorithm initiating with the nodes labeled "European" and "Huguenots", the PPR algorithm struggles to find the appropriate subgraphs around them that define the most related passage. This occurs when multiple passages exist in the corpus that discuss very similar topics since the PPR algorithm is not able to leverage query context directly.

Table 11: Error analysis on MuSiQue.

| Error Type | Error Percentage (%) |
|---|---|
| NER Limitation | 48 |
| Incorrect/Missing OpenIE | 28 |
| PPR | 24 |

## F.2 Concepts vs. Context Tradeoff

Given our method's entity-centric nature in extraction and indexing, it has a strong bias towards concepts that leaves many contextual signals unused. This design enables single-step multi-hop retrieval while also enabling contextual cues to avoid distracting from more salient entities. As seen in the first example in Table 12, ColBERTv2 uses the context to retrieve passages that are related to famous Spanish navigators but not "Sergio Villanueva", who is a boxer. In contrast, HippoRAG is able to hone in on "Sergio" and retrieve one relevant passage.

Unfortunately, this design is also one of our method's greatest limitations since ignoring contextual cues accounts for around 48% of errors in our small-scale error analysis. This problem is more apparent in the second example since the concepts are general, making the context more important. Since the only concept tagged by HippoRAG is "protons", it extracts passages related to "Uranium" and "nuclear weapons" while ColBERTv2 uses the context to extract more relevant passages associated with the discovery of atomic numbers.

Table 12: Examples showing the concept-context tradeoff on MuSiQue.

| Question | HippoRAG | ColBERTv2 |
|---|---|---|
| Whose father was a navigator who explored the east coast of the continental region where **Sergio Villanueva** would later be born? | **Sergio Villanueva** 
 César Gaytan 
 Faustino Reyes | Francisco de Eliza (navigator) 
 Exploration of N. America 
 Vicente Pinzón (navigator) |
| What undertaking included the person who discovered that the number of **protons** in each element's atoms is unique? | Uranium 
 Chemical element 
 History of nuclear weapons | **Atomic number** 
 Atomic theory 
 Atomic nucleus |

Table 13: **Single-step retrieval performance.** HippoRAG performs substantially better on MuSiQue and 2WikiMultiHopQA than all baselines and achieves comparable performance on the less challenging HotpotQA dataset.

| Model | Retriever | MuSiQue | | 2Wiki | | HotpotQA | | Average | |
|---|---|---|---|---|---|---|---|---|---|
| | | R@2 | R@5 | R@2 | R@5 | R@2 | R@5 | R@2 | R@5 |
| Baseline | Contriever | 34.8 | 46.6 | 46.6 | 57.5 | 57.2 | 75.5 | 46.2 | 59.9 |
| | ColBERTv2 | 37.9 | 49.2 | 59.2 | 68.2 | **64.7** | 79.3 | 53.9 | 65.6 |
| HippoRAG | Contriever | 41.0 | 52.1 | 71.5 | **89.5** | 59.0 | 76.2 | 57.2 | 72.6 |
| | ColBERTv2 | 40.9 | 51.9 | 70.7 | 89.1 | 60.5 | 77.7 | 57.4 | 72.9 |
| HippoRAG w/ | Contriever | 42.3 | 54.5 | 71.3 | 87.2 | 60.6 | 79.1 | 58.1 | 73.6 |
| Uncertainty Ensemble | ColBERTv2 | **42.5** | **54.8** | **71.9** | 89.0 | 62.5 | **80.0** | **59.0** | **74.6** |

To get a better trade-off between concepts and context, we introduce an ensembling setting where HippoRAG scores are ensembled with dense retrievers when our parahippocampal region shows uncertainty regarding the link between query and KG entities. This process represents instances when no hippocampal index was fully activated by the upstream parahippocampal signal and thus the neocortex must be relied on more strongly. We only use uncertainty ensembling if one of the query-KG entity scores $cosine\_similarity(M(c_i), M(e_j))$ is lower than a threshold $\theta$, for example, if there was no *Stanford* node in the KG and the closest node in the KG is something that has a cosine similarity lower than $\theta$ such as *Stanford Medical Center*. The final passage score for uncertainty

ensembling is the average of the HippoRAG scores and standard passage retrieval using model $M$, both of which are first normalized into the 0 to 1 over all passages.

When HippoRAG is ensembled with $M$ under *"Uncertainty Ensemble"*, it further improves on MuSiQue and outperforms our baselines in R@5 for HotpotQA, as shown in Table 13. When used in combination with IRCoT, as shown in Table 14, the ColBERTv2 ensemble outperforms all previous baselines in both R@2 and R@5 on HotpotQA. Although the simplicity of this approach is promising, more work needs to be done to solve this context-context tradeoff since simple ensembling does lower performance in some cases, especially for the 2WikiMultiHopQA dataset.

Table 14: **Multi-step retrieval performance.** Combining HippoRAG with standard multi-step retrieval methods like IRCoT results in substantial improvements on all three datasets.

| Model | Retriever | MuSiQue | | 2Wiki | | HotpotQA | | Average | |
|---|---|---|---|---|---|---|---|---|---|
| | | R@2 | R@5 | R@2 | R@5 | R@2 | R@5 | R@2 | R@5 |
| IRCoT | Contriever | 39.1 | 52.2 | 51.6 | 63.8 | 65.9 | 81.6 | 52.2 | 65.9 |
| | ColBERTv2 | 41.7 | 53.7 | 64.1 | 74.4 | 67.9 | 82.0 | 57.9 | 70.0 |
| IRCoT + HippoRAG | Contriever | 43.9 | 56.6 | 75.3 | 93.4 | 65.8 | 82.3 | 61.7 | 77.4 |
| | ColBERTv2 | **45.3** | 57.6 | **75.8** | **93.9** | 67.0 | 83.0 | **62.7** | 78.2 |
| IRCoT + HippoRAG w/ | Contriever | 44.4 | **58.5** | 75.3 | 91.5 | 66.9 | 85.0 | 62.2 | **78.3** |
| Uncertainty Ensemble | ColBERTv2 | 40.2 | 53.4 | 74.5 | 91.2 | **68.2** | **85.3** | 61.0 | 76.6 |

## F.3 OpenIE Limitations

OpenIE is a critical step in extracting structured knowledge from unstructured text. Nonetheless, its shortcomings can result in gaps in knowledge that may impair retrieval and QA capabilities. As shown in Table 15, GPT-3.5 Turbo overlooks the crucial song title "Don't Let Me Wait Too Long" during the OpenIE process. This title represents the most significant element within the passage. A probable reason is that the model is insensitive to such a long entity. Besides, the model does not accurately capture the beginning and ending years of the war, which are essential for the query. This is an example of how models routinely ignore temporal properties. Overall, these failures highlight the need to improve the extraction of critical information.

Table 15: Open information extraction error examples on MuSiQue.

| Question | Passage | Missed Triples |
|---|---|---|
| What company is the label responsible for "Don't Let Me Wait Too Long" a part of? | "Don't Let Me Wait Too Long" was sequenced on side one of the LP, between the ballads "The Light That Has Lighted the World" and "Who Can See It" ... | (Don't Let Me Wait Too Long, sequenced on, side one of the LP) |
| When did the president of the Confederate States of America end his fight in the Mexican-American war? | Jefferson Davis fought in the Mexican–American War (1846–1848), as the colonel of a volunteer regiment ... | (Mexican-American War, starts, 1846), (Mexican-American War, ends, 1848) |

## F.4 OpenIE Document Length Analysis

Finally, we present a small-scale intrinsic experiment to help us understand the robustness of our OpenIE methods to increasing passage length. The length-dependent evaluation results in Table 16, show that GPT-3.5-Turbo OpenIE results deteriorate substantially when extracting from longer instead of shorter passages. This is likely due to a higher sentence and paragraph complexity for longer passages which leads to lower quality extraction. More work is needed to address this limitation since further chunking would only create other issues due to sentence interdependence.

Table 16: Intrinsic OpenIE evaluation using the CaRB [6] framework. Performance difference between the 10 longest and 10 shortest annotated passages using our default GPT-3.5 Turbo (1106) model.

|  | AUC | Precision | Recall | F1 |
|---|---|---|---|---|
| 10 Shortest Passages | 58.9 | 79.2 | 65.7 | 71.8 |
| 10 Longest Passages | 39.0 | 60.7 | 48.5 | 53.9 |

# G    Cost and Efficiency Comparison

One of HippoRAG's main advantages against iterative retrieval methods is the dramatic online retrieval efficiency gains brought on by its single-step multi-hop retrieval ability in terms of both cost and time. Specifically, as seen in Table 17, retrieval costs for IRCoT are 10 to 30 times higher than HippoRAG since it only requires extracting relevant named entities from the query instead of processing all of the retrieved documents. In systems with extremely high usage, a cost difference of an order of magnitude such as this one could be extremely important. The difference with IRCoT in terms of latency is also substantial, although more challenging to measure exactly. Also as seen in Table 17, HippoRAG can be 6 to 13 times faster than IRCoT, depending on the number of retrieval rounds that need to be executed (2-4 in our experiments).[6]

Table 17: Average cost and efficiency measurements for online retrieval using GPT-3.5 Turbo on 1,000 queries.

|  | ColBERTv2 | IRCoT | HippoRAG |
|---|---|---|---|
| API Cost ($) | 0 | 1-3 | 0.1 |
| Time (minutes) | 1 | 20-40 | 3 |

Although offline indexing time and costs are higher for HippoRAG than IRCoT—around 10 times slower and $15 more expensive for every 10,000 passages [7], these costs can be dramatically reduced by leveraging open source LLMs. As shown in our ablation study in Table 5 Llama-3.1-70B-Instruct [1] performs similarly to GPT-3.5 Turbo even though it can be deployed locally using vLLM [40] and 4 H100 GPUs to index 10,000 documents in around 4 hours, as seen in Table 18. Additionally, since these costs could be even further reduced by locally deploying this model, the barriers for using HippoRAG at scale could be well within the computational budget of many organizations. Finally, we note that even if LLM generation cost drops, the online retrieval efficiency gains discussed above remain intact given that the number of tokens required for IRCoT vs. HippoRAG stay constant and LLM use is likely to also remain the system's main computational bottleneck.

Table 18: Average cost and latency measurements for offline indexing using GPT-3.5 Turbo and locally deployed Llama-3.1 (8B and 70B) using vLLM on 10,000 passages.

| Model | Metric | ColBERTv2 | IRCoT | HippoRAG |
|---|---|---|---|---|
| GPT-3.5 Turbo-1106 (Main Results) | API Cost ($) | 0 | 0 | 15 |
|  | Time (minutes) | 7 | 7 | 60 |
| GPT-3.5 Turbo-0125 | API Cost ($) | 0 | 0 | 8 |
|  | Time (minutes) | 7 | 7 | 60 |
| Llama-3.1-8B-Instruct | API Cost ($) | 0 | 0 | 0 |
|  | Time (minutes) | 7 | 7 | 120 |
| Llama-3.1-70B-Instruct | API Cost ($) | 0 | 0 | 0 |
|  | Time (minutes) | 7 | 7 | 250 |

---

[6]We use a single thread to query the OpenAI API for online retrieval in both IRCoT and HippoRAG. Since IRCoT is an iterative process and each of the iterations must be done sequentially, these speed comparisons are appropriate.

[7]To speed up indexing, we use 10 threads querying *gpt-3.5-turbo-1106* through the OpenAI API in parallel. At the time of writing, the cost of the API is $1 for a million input tokens and $2 for a million output tokens.

# H   Implementation Details & Compute Requirements

Apart from the details included in §3.4, we use implementations based on PyTorch [59] and HuggingFace [86] for both Contriever [35] and ColBERTv2 [70]. We use the python-igraph [13] implementation of the PPR algorithm. For BM25, we employ Elastic Search [24]. For multi-step retrieval, we use the same prompt implementation as IRCoT [78] and retrieve the top-10 passages at each step. We set the maximum number of reasoning steps to 2 for HotpotQA and 2WikiMultiHopQA and 4 for MuSiQue due to their maximum reasoning chain length. We combine IRCoT with different retrievers by replacing its base retriever BM25 with each retrieval method, including HippoRAG, noted as "IRCoT + HippoRAG" below.[8] For the QA reader, we use top-5 retrieved passages as the context and 1-shot QA demonstration with CoT prompting strategy [78].

In terms of compute requirements, most of our compute requirements are unfortunately not disclosed by the OpenAI. We run ColBERTv2 and Contriever for indexing and retrieval we use 4 NVIDIA RTX A6000 GPUs with 48GB of memory. For indexing with Llama-3.1 models, we use 4 NVIDIA H100 GPUs with 80GB of memory. Finally, we used 2 AMD EPYC 7513 32-Core Processors to run the Personalized PageRank algorithm.

# I   LLM Prompts

The prompts we used for indexing and query NER are shown in Figure 7 and Figure 8, while the OpenIE prompt is shown in Figure 9.

---

**Passage NER (Indexing)**

**Instruction:**

Your task is to extract named entities from the given paragraph.
Respond with a JSON list of entities.

**One-Shot Demonstration:**

Paragraph:
```

Radio City
Radio City is India's first private FM radio station and was started on 3 July 2001. It plays Hindi, English and regional songs. Radio City recently forayed into New Media in May 2008 with the launch of a music portal - PlanetRadiocity.com that offers music related news, videos, songs, and other music-related features.
```

{"named_entities": ["Radio City", "India", "3 July 2001", "Hindi","English", "May 2008", "PlanetRadiocity.com"]}

**Input:**

Paragraph:
```
PASSAGE TO INDEX
```

---

Figure 7: Prompt for passage NER during indexing.

---

[8]Since the original IRCoT does not provide a score for each retrieved passage, we employ beam search for the iterative retrieval process. Each candidate passage maintains the highest historical score during beam search.

## Query NER (Retrieval)

**Instruction:**

You're a very effective entity extraction system. Please extract all named entities that are important for solving the questions below. Place the named entities in JSON format.

**One-Shot Demonstration:**

Question: Which magazine was started first Arthur's Magazine or First for Women?

{"named_entities": ["First for Women", "Arthur's Magazine"]}

**Input:**

Question: **QUERY TO INDEX**

Figure 8: Prompt for query NER during retrieval.

## Open Information Extraction

**Instruction:**

Your task is to construct an RDF (Resource Description Framework) graph from the given passages and named entity lists.
Respond with a JSON list of triples, with each triple representing a relationship in the RDF graph.
Pay attention to the following requirements:
- Each triple should contain at least one, but preferably two, of the named entities in the list for each passage.
- Clearly resolve pronouns to their specific names to maintain clarity.

Convert the paragraph into a JSON dict, it has a named entity list and a triple list.

**One-Shot Demonstration:**

Paragraph:
```
Radio City
Radio City is India's first private FM radio station and was started on 3 July 2001. It plays Hindi, English and regional songs. Radio City recently forayed into New Media in May 2008 with the launch of a music portal - PlanetRadiocity.com that offers music related news, videos, songs, and other music-related features.
```
{"named_entities": ["Radio City", "India", "3 July 2001", "Hindi","English", "May 2008", "PlanetRadiocity.com"]}

{"triples":
  [
    ["Radio City", "located in", "India"],
    ["Radio City", "is", "private FM radio station"],
    ["Radio City", "started on", "3 July 2001"],
    ["Radio City", "plays songs in", "Hindi"],
    ["Radio City", "plays songs in", "English"],
    ["Radio City", "forayed into", "New Media"],
    ["Radio City", "launched", "PlanetRadiocity.com"],
    ["PlanetRadiocity.com", "launched in", "May 2008"],
    ["PlanetRadiocity.com", "is", "music portal"],
    ["PlanetRadiocity.com", "offers", "news"],
    ["PlanetRadiocity.com", "offers", "videos"],
    ["PlanetRadiocity.com", "offers", "songs"]
  ]
}

**Input:**

Convert the paragraph into a JSON dict, it has a named entity list and a triple list.
Paragraph:
```
**PASSAGE TO INDEX**
```
{"named_entities": [**NER LIST**]}

Figure 9: Prompt for OpenIE during indexing.

