# OpenReview forum: "HippoRAG: Neurobiologically Inspired Long-Term Memory for Large Language Models"
_NeurIPS.cc/2024/Conference — NeurIPS 2024 poster_

### Official Review · Reviewer_oPeR · 2024-07-04

**Soundness:** 3
**Presentation:** 3
**Contribution:** 3
**Rating:** 7
**Confidence:** 4

**Summary:**

This paper proposes a retrieval-augmented generation (RAG) method that is inspired by the hippocampal indexing theory of human memory to enable longer knowledge storage and efficient knowledge integration over new experiences.

**Strengths:**

1. This paper's idea is interesting and shows impressive performances.
2. The paper is well-written and the presentation is clear.
3. The metric `Node Specificity` is aligned with the intuition that humans may get better memorization of things that they are seeing over and over again. This is shown in Figure 2 where the logo of "Stanford" grows larger. I think this one is very interesting.

**Weaknesses:**

1. Extracting triplets from the passage strongly depends on the power of the triplets extracting model, which is a language model in the paper's implementation. I'm concerned that it may lose information when the passage becomes longer. Is this a common strategy in other retrieval methods?
2. Missing citations. Since the paper talks about long-term memory in related work, I believe the following papers may need to be cited.
[1] Memoria: Resolving Fateful Forgetting Problem through Human-Inspired Memory Architecture.
[2] MEMORYLLM: Towards Self-Updatable Large Language Models.
[3] CAMELoT: Towards Large Language Models with Training-Free Consolidated Associative Memory.

**Questions:**

see weaknesses

**Limitations:**

yes, the authors have addressed the limitations.

---

> ### Author Rebuttal · Authors · 2024-08-07
>
> We thank the reviewer for the time and effort spent reviewing our paper. We are glad they found our method interesting and enjoyed the `Node Specificity` portion of our methodology. We also appreciate their suggestions and long-term memory references, which we will definitely include in our updated literature review.
>
> - **W1: Extracting triplets from the passage strongly depends on the power of the triplets extracting model, which is a language model in the paper's implementation. I'm concerned that it may lose information when the passage becomes longer. Is this a common strategy in other retrieval methods?**
>
> We refer the reviewer to the general response for an intrinsic and extrinsic length-dependent evaluation of our knowledge graph construction.

---

> > ### Comment · Reviewer_oPeR · 2024-08-13
> > **Response to the rebuttal**
> >
> > Thank the authors for the responses. I don't have more concerns.

---

### Official Review · Reviewer_bGvw · 2024-07-10

**Soundness:** 3
**Presentation:** 3
**Contribution:** 2
**Rating:** 5
**Confidence:** 4

**Summary:**

This paper presents Hippo-RAG, which enables knowledge integration across retrieval results and supports long-term memory with a mechanism that resembles the hippocampal memory indexing theory. Hippo-RAG includes two steps: offline indexing to extract, encode, and index the passages to KG, and online retrieval to extract entities from the query and retrieve the results from the KG. Hippo-RAG outperforms various existing methods on the multi-hop QA benchmarks and is able to perform multi-hop retrieval in a single step.

**Strengths:**

1. The proposed method is novel, and it resembles the long-term memory mechanism of human beings. The offline indexing stage enables knowledge integration across passages and addresses the path-finding problems in RAG. The pipeline examples can help to better understand the process.
2. The proposed method achieves significant performance improvement from various challenging multi-hop QA benchmarks.
3. The paper provides sufficient details for reproduction.

**Weaknesses:**

1. The mechanism is great, but the generalization of the method is not good enough. Intuitively, the mentioned mechanism should work for several scenarios that require long-term memory. Still, the proposed method narrows it down and limits the experiments to multi-hop reasoning.
2. The framework's performance depends on the NER ability of LLM. However, [1] demonstrates a large performance gap between prompting LLMs for NER and fine-tuned NER models. The error analysis also indicates that most errors in the system are from NER.
3. The comparison of baselines may not be fair since the parameter sizes of Hippo-RAG are significantly larger than other methods. Contriever and ColBERTv2 achieve comparable results on MuSiQue and HotpotQA, but when combined with HippoRAG, the improvement is insignificant. ColBERTv2 is the second-best model, and the improvement from Hippo-RAG seems incremental on the two datasets.



[1] https://arxiv.org/abs/2304.10428

**Questions:**

1. Can Hippo-RAG work for other tasks that require long-term memory?
2. Why does Hippo-RAG perform significantly better on 2Wiki?

**Limitations:**

The authors adequately discussed the limitations and potential negative societal impact of their work.

---

> ### Author Rebuttal · Authors · 2024-08-07
>
> We sincerely appreciate the reviewer for the time and effort they dedicated to reviewing our paper as well as for their comments and questions.
>
> - **W1: The mechanism is great, but the generalization of the method is not good enough. Intuitively, the mentioned mechanism should work for several scenarios that require long-term memory. Still, the proposed method narrows it down and limits the experiments to multi-hop reasoning.**
> - **Q1: Can Hippo-RAG work for other tasks that require long-term memory?**
>
> Long-term memory in humans is a remarkably complex and powerful cognitive faculty that forms the basis for our reasoning and decision making. It’s a holy grail for artificial intelligence to acquire a memory mechanism as powerful as humans’, but that’s bound to be a long journey that we as a community have to take, one step at a time. Before HippoRAG, the de facto solution for long-term memory for LLMs, that expose a static LLM to new experiences, is RAG (see, e.g., this blog post that has had important impact in shaping this belief and practice: https://lilianweng.github.io/posts/2023-06-23-agent/). However, current RAG lacks many important properties of human long-term memory such as the inability to store previously learned procedures and store facts associated only with a particular time or place (episodic memory). HippoRAG focuses on another one of these important properties that current RAG lacks, knowledge integration across experiences, for which the hippocampus is believed to play an important role. For this property, multi-hop QA is the most natural task for evaluation because of its inherent knowledge integration requirement as well as the well established datasets and baselines.
>
> Can HippoRAG be applied to other tasks that require long-term memory, e.g., agents that need to maintain episodic or procedural memory? Maybe, but more likely new adaptations and innovations will be needed to properly handle new properties of such memories, e.g., handling of their temporospatial attributes. We are very interested in further extending HippoRAG to get even closer to the powerful human long-term memory and handle more tasks that require long-term memory. On the other hand, we also believe that HippoRAG still marks a meaningful and solid step in the long journey of trying to empower AI with the powerful long-term memory mechanism with which humans are equipped.
>
> - **W2: The framework's performance depends on the NER ability of LLM. However, [1] demonstrates a large performance gap between prompting LLMs for NER and fine-tuned NER models. The error analysis also indicates that most errors in the system are from NER.**
>
> We agree with the reviewer that understanding the effect of different NER components on HippoRAG’s performance is important. In order to do this, we compare our standard query NER prompting method with the state-of-the-art NER model UniversalNER [1] below:
>
> | MuSiQue Retrieval                          | R@2  | R@5  |
> | ------------------------------------------ | ---- | ---- |
> | HippoRAG (Contriever) w/ UniversalNER-7B   | 25.1 | 32.0 |
> | HippoRAG (Contriever) w/ GPT-3.5-turbo NER | 41.0 | 52.1 |
>
> We can see that GPT-3.5-turbo outperforms UniversalNER-7B on MuSiQue. We believe this is due to UniversalNER’s preference for a few entity types given its training data. As reported in their paper, the top 1% most frequent entity types account for 74% of all entities produced. Additionally, even though UniversalNER is one of the most flexible supervised NER models available, it still requires defining entity types a priori, in contrast to our prompting methodology.
> Nevertheless, as the reviewer mentioned, we acknowledge that query NER issues are still HippoRAG’s largest source of errors. However, as we discuss in Appendix F.1 and F.2, these errors do not come from NER issues per se but rather the use of NER exclusively to link queries to our KG. For example, in a query such as “When was one internet browser’s version of Windows 8 made accessible?”, the terms that must be extracted are not named entities but common noun phrases like “internet browser”. We believe that finding suitable starting points for graph search is an important venue for future research.

---

> > ### Comment · Reviewer_bGvw · 2024-08-11
> >
> > Thank you for the analyses provided. I raise the rating from 4 to 5 for they address most of my concerns.
> >
> > My remaining concerns are about W3 experimental comparison.
> > The settings for RAPTOR are not stated in the paper (which retriever (SBERT/DPR/BM25) is used for its implementation). In Table 2, ColBERTv2 outperforms the two LLM-augmented methods, and it is not discussed in the paper. These are my concerns, but they are not the weakness of the paper. I would appreciate it if the authors could include them in later versions.

---

> > > ### Author Response · Authors · 2024-08-12
> > >
> > > Thank you for considering our responses and kindly updating your score. Please feel free to let us know if you have any further questions or concerns.
> > >
> > > We appreciate your suggestion to include more details about the LLM-augmented baselines. In order to ensure our experiments appropriately represent these baselines, we used the embedding methods which obtain the strongest performance in their respective experiments. Both of these were Sentence Transformer models, `sentence-transformers/multi-qa-mpnet-base-cos-v1` for RAPTOR and GTR (`sentence-transformers/gtr-t5-base`) for the Proposition-izer.
> > >
> > > In order to provide a direct comparison with ColBERTv2 as the reviewer suggests, we ran these two baselines using ColBERTv2 as their retrieval component.
> > >
> > > | Method                    | MuSiQue  | MuSiQue  | 2Wiki   | 2Wiki   | HotpotQA | HotpotQA |
> > > |---------------------------|----------|----------|---------|---------|----------|----------|
> > > |                           | R@2      | R@5      | R@2     | R@5     | R@2      | R@5      |
> > > | ColBERTv2                 | 37.9     | 49.2     | 59.2    | 68.2    | **64.7**     | **79.3**     |
> > > | RAPTOR                    | 35.7     | 45.3     | 46.3    | 53.8    | 58.1     | 71.2     |
> > > | RAPTOR (ColBERTv2)        | 36.9     | 46.5     | 57.3    | 64.7    | 63.1     | 75.6     |
> > > | Proposition               | 37.6     | 49.3     | 56.4    | 63.1    | 58.7     | 71.1     |
> > > | Proposition (ColBERTv2)   | 37.8     | 50.1     | 55.9    | 64.9    | 63.9     | 78.1     |
> > > | HippoRAG (ColBERTv2)      | **40.9** | **51.9** | **70.7**| **89.1**| 60.5 | 77.7 |
> > >
> > > We find that both baselines obtain stronger performance using ColBERTv2 than their original models. However, they still underperform ColBERTv2 itself in all results except for MuSiQue (R@5), where only the Proposition-izer model outperforms ColBERTv2 and gets 0.8% closer to HippoRAG.
> > >
> > > We point out that RAPTOR's poor performance compared to ColBERTv2 demonstrates that their "cluster and summarize" methodology is mostly ineffective for the comprehensive knowledge integration required in these datasets. Finally, we note that the Proposition-izer's  unstable performance, especially its poor performance in 2Wiki, illustrates that separating passages based on propositions can sharply diminish a retriever's knowledge integration capabilities.
> > >
> > > We will include these results and discussion in the revised version.

---

> ### Author Response · Authors · 2024-08-07
> **Rebuttal by Authors (Continued)**
>
> - **W3: The comparison of baselines may not be fair since the parameter sizes of Hippo-RAG are significantly larger than other methods. Contriever and ColBERTv2 achieve comparable results on MuSiQue and HotpotQA, but when combined with HippoRAG, the improvement is insignificant. ColBERTv2 is the second-best model, and the improvement from Hippo-RAG seems incremental on the two datasets.**
>
> In the response below, we present evidence and arguments that we hope will convince the reviewer that our performance improvements are substantial and our experimental setting is sound and fair.
>
> #### **Improvements on MuSiQue**
>
> We first highlight that our method “achieves significant performance improvement from various challenging multi-hop QA benchmarks”, as pointed out by the reviewer when listing our strengths.
> HippoRAG improves R@5 on MuSiQue by 5.5% and 2.7% over Contriever and ColBERTv2 respectively, as well as 3.4% F1 score on QA compared to ColBERTv2. Many well-received and highly cited works such as IRCoT [2], Self-Ask [3], ITER-RETGEN [4] and MCR [5] highlight improvements similar to ours.
>
> #### **HotpotQA’s Weaknesses as a Knowledge-Integration Benchmark**
>
> As discussed in subsection “Single-Step Retrieval Results” (Section 4) of our paper, our lower performance on HotpotQA is mainly due to its lower need for knowledge integration due to existing shortcut signals. This HotpotQA limitation is referenced when constructing MuSiQue [6], but also explored more deeply in Appendix B of our paper. In bridge multi-hop questions, the query and second supporting passage should be linked only through a bridge entity. However, as shown in Figure 6, HotpotQA queries are on average as similar to the second supporting passage as to their distractors, instead of less similar like in the other datasets, making that second supporting document easier to detect without knowledge integration.
>
> #### **Measuring the Impact of Improved Knowledge-Integration Directly**
>
> Additionally, when we measure the impact of improved knowledge integration in MuSiQue and 2WikiMultiHopQA, our improvements are even larger than our overall results. In Table 8, we show that HippoRAG’s ability to find all supporting documents is key to its strong performance, obtaining a 6.3% and 38.6% improvement over ColBERTv2 in All-Recall@5, compared to a 2.7% and 21.3% improvement in standard Recall@5.
>
> #### **Our Experimental Setting is Sound and Fair**
>
> Finally, we refer to the reviewer’s concern that our comparison with Contriever and ColBERTv2 might not be fair given their small size compared to LLM. We understand the reviewer’s apprehension, however, we argue that improving the performance of a system using an LLM and comparing it to the original system is a well-accepted paradigm in AI research. As a specific example, many well-received QA works such as IRCoT [2], Self-Ask [3] and RAPTOR [7] leverage a large LLM in different ways and compare with the original pipeline; which in these papers contain various retrieval methods like BM25, DPR [8] and ColBERTv2. We note that HippoRAG follows this same research paradigm, which is related to the large and growing body of recent work that leverages LLMs to challenge prior state of the art based on smaller models.
> As a final remark, we also note that our method outperforms RAPTOR and is comparable with IRCoT (and considerably more efficient). Since both of these methods augment the original retrieval process with outputs from an LLM, they can be seen as HippoRAG’s most direct baselines in that sense.
>
> - **Q2: Why does Hippo-RAG perform significantly better on 2Wiki?**
>
> As discussed in subsection “Single-Step Retrieval Results” (Section 4) of our paper, 2WikiMultiHopQA’s construction is more entity-centric than the other two datasets, making it particularly well-suited for HippoRAG’s design.
>
> To provide some extra context, 2WikiMultiHopQA was created by leveraging the Wikidata KG to determine which entities and relations could be found in Wikipedia passages to create compositional, comparison, inference or bridge-comparison multi-hop questions. Due to this construction process, queries in this dataset include at least one named entity, a characteristic that our methodology can leverage given its previously mentioned reliance on NER to link queries to the KG.

---

> ### Author Response · Authors · 2024-08-07
> **Rebuttal by Authors (Continued)**
>
> ### References
>
> [1] Zhou et al. (2024). UniversalNER: Targeted Distillation from Large Language Models for Open Named Entity Recognition.\
> [2] Trivedi et al. (2023). Interleaving Retrieval with Chain-of-Thought Reasoning for Knowledge-Intensive Multi-Step Questions.\
> [3] Press et al. (2023). Measuring and Narrowing the Compositionality Gap in Language Models.\
> [4] Shao et al. (2023). Enhancing Retrieval-Augmented Large Language Models with Iterative Retrieval-Generation Synergy.\
> [5] Yoran et al. (2023). Answering Questions by Meta-Reasoning over Multiple Chains of Thought.\
> [6] Trivedi et al. (2022). ♫ MuSiQue: Multihop Questions via Single-hop Question Composition.\
> [7] Sarthi et al. (2024). RAPTOR: Recursive Abstractive Processing for Tree-Organized Retrieval.\
> [8] Karpukhin et al. (2020). Dense Passage Retrieval for Open-Domain Question Answering.

---

### Official Review · Reviewer_fKDL · 2024-07-13

**Soundness:** 3
**Presentation:** 4
**Contribution:** 3
**Rating:** 7
**Confidence:** 4

**Summary:**

The paper introduces HippoRAG, a retrieval framework inspired by hippocampal indexing theory to enhance large language models (LLMs) in integrating new information. The algorithm is a combination of LLMs, knowledge graphs (KGs), and the Personalized PageRank algorithm. HippoRAG outperforms existing retrieval-augmented generation (RAG) methods in multi-hop question answering (QA) by a significant margin. It achieves comparable or better performance than iterative retrieval methods like IRCoT while being significantly cheaper and faster.

**Strengths:**

- The idea of converting the corpus to KG and then running page rank algorithm for better retrievals is interesting. The connections to hippocampal memory theory is inspiring.

- The method is shown to be more efficient than iterative retrieval method IRCoT and can also use IRCoT to further boost the performance.

- The experiment is well-executed, showing the improvements of HippoRAG.

- The paper is well-written and easy to follow. Analysis is clear.

**Weaknesses:**

- Some baselines from KG-LLM for multi-hop QA literature are missing. These could be beneficial to replace Page Rank for ablation study or comparing with KGQA on open-source knowledge graph.

- Some citation are missing (see questions)

**Questions:**

- Can we evaluate the performance of constructed knowledge graph alone?

- Some citations are missing:

    - Park et al. Graph Elicitation for Guiding Multi-Step Reasoning in Large Language Models

    - Jin et al. Improving embedded knowledge graph multi-hop question answering by introducing relational chain reasoning

**Limitations:**

Yes

---

> ### Author Rebuttal · Authors · 2024-08-07
>
> We thank the reviewer for the time and effort spent on reviewing our paper. We appreciate their helpful suggestions and the related work they brought to our attention.
>
> - **W1: Some baselines from KG-LLM for multi-hop QA literature are missing. These could be beneficial to replace Page Rank for ablation study or comparing with KGQA on open-source
> knowledge graph.**
>
> In order to address this important concern, we discuss previous works on multi-hop QA that combine graphs and neural methods by dividing them in two major categories 1) graph-augmented reading comprehension and 2) graph-augmented retrieval.
>
> #### **Graph-Augmented Reading Comprehension**
>
> In this direction, graphs are used to provide structure to complement textual signals from the retrieved passages and improve QA performance.
>
> Most supervised methods in this category [1,2,3], train a graph neural network (GNN) to introduce hyperlinks or co-occurrence graphs into a language model for better QA. For works using LLM prompting, the graphs are usually constructed by extracting triples from retrieved passages and then added to the final LLM prompt [4,5,6]. The contributions from these works are mostly orthogonal to our own since graphs are not used in the retrieval process like they are in HippoRAG, however, they could be used in conjunction with our method to achieve complementary improvements.
>
> #### **Graph-Augmented Retrieval**
>
> In the second category, the graph is used to retrieve relevant documents rather than provide structured signals from a previously retrieved set. Many previous such works train a re-ranker to traverse a graph primarily made from hyperlinks [7,8,9,10,11,12].
> As far as we know, HippoRAG is the first method that successfully combines a KG and an LLM for retrieval in multi-hop QA without any supervised data or predefined Wikipedia hyperlinks, meaning it can be used in more scenarios than previous methods.
>
> We hope that this discussion helps better contextualize our method and clarify its uniqueness. As discussed above, although much KG-LLM work has been done for multi-hop QA, most of it is either not used in the retrieval process or relies on supervised data. However, as the reviewer mentioned, some KGQA methods are able to leverage an open KG for question answering using LLM prompting.  For completeness, we carried out an ablation of our method that leverages our KG using Pangu (Gu et al 2023), a state-of-the-art LLM-based KBQA method, for QA.
>
> Pangu obtains less than 0.05% F1 score on MuSiQue, demonstrating that standard KGQA systems cannot perform question answering on our OpenIE KG. We observe that the following challenges and many more lead to this dramatically low performance:
>
> #### Entity Linking
>
> This is a significant challenge even in settings with strictly defined KG, making it even more difficult on our KG where one entity can have several expressions. For example, for the question "When was the person who Messi's goals in Copa del Rey compared to get signed by Barcelona?", many entities are potentially relevant, such as Messi, the Spanish Messi, or Lionel Messi. This diversity makes it difficult to identify which starting point could lead to the answer.
>
> #### OpenIE Coverage of Answers and Relations
>
> Some relations as well as final QA answers are sometimes not present in the KG even if they are in the text. For instance, as we demonstrated in Table 14, this is the case for some complex expressions and numerical attributes.
>
> We will expand our discussion of this literature accordingly in the revised version.
>
> - **Q1: Can we evaluate the performance of constructed knowledge graph alone?**
>
> We refer the reviewer to the general response for an intrinsic evaluation of our knowledge graph construction.
>
> ### References
>
> [1] Fang et al. (2020). Hierarchical Graph Network for Multi-hop Question Answering.\
> [2] Ramesh et al. (2023). Single Sequence Prediction over Reasoning Graphs for Multi-hop QA.\
> [3] Qiu et al. (2019). Dynamically Fused Graph Network for Multi-hop Reasoning.\
> [4] Park et al. (2023). Graph Elicitation for Guiding Multi-Step Reasoning in Large Language Models.\
> [5] Li et al. (2023). Leveraging structured information for explainable multi-hop question answering and reasoning.\
> [6] Liu et al. (2024). Era-cot: Improving chain-of-thought through entity relationship analysis.\
> [7] Ding et al. (2019). Cognitive Graph for Multi-Hop Reading Comprehension at Scale.\
> [8] Zhu et al. (2021). Adaptive Information Seeking for Open-Domain Question Answering.\
> [9] Nie et al. (2019). Revealing the Importance of Semantic Retrieval for Machine Reading at Scale.\
> [10] Das et al. (2019). Multi-step Entity-centric Information Retrieval for Multi-Hop Question Answering.\
> [11] Asai et al. (2020). Learning to retrieve reasoning paths over wikipedia graph for question answering.\
> [12] Li et al. (2021). Hopretriever: Retrieve hops over wikipedia 3625 to answer complex questions.\
> [13] Gu et al. (2023). Don’t Generate, Discriminate: A Proposal for Grounding Language Models to Real-World Environments.

---

> > ### Comment · Reviewer_fKDL · 2024-08-10
> > **Thank you!**
> >
> > Thank the authors for their detailed rebuttal. My concerns are addressed adequately and I will keep my score.

---

> > > ### Author Response · Authors · 2024-08-12
> > >
> > > Thank you again for your helpful suggestions and for considering our response.

---

### Author Rebuttal · Authors · 2024-08-07

We thank all of the reviewers for the time and effort they dedicated to reviewing our work, we believe our work will be significantly enhanced from incorporating their suggestions.

We are delighted to know that reviewers found the parallels between our methodology and hippocampal memory indexing theory inspiring. We are also happy that our proposed method was not only deemed interesting and novel but also recognized achieving impressive performance while being efficient. Lastly, we appreciate the reviewers’ comments on the clarity of our writing, experiments and analysis.

### OpenIE Performance (R1 & R3)

We appreciate both reviewers’ questions concerning the intrinsic performance of our OpenIE methodology as well as its robustness to longer documents. We will address these related concerns below:

- **(R1-Q1): Can we evaluate the performance of constructed knowledge graph alone?**

We first note that our ideal IE output is quite different from that of the conventional ClosedIE or OpenIE settings, which is too constrained and unconstrained respectively in terms of named entities and pre-defined relations. Therefore, to evaluate our KGC performance intrinsically, we extract OpenIE triples manually from a small dataset of 20 passages taken from the MuSiQue training set. Out of these passages, we extract 239 gold triples. We measure the quality of our KGC using the CaRB [1] metrics and compare different LLMs with the supervised closed information extraction model REBEL [2] and IMoJIE [3], a supervised OpenIE method.

| Model               |   AUC | Precision | Recall |    F1 |
|---------------------|-------|-----------|--------|-------|
| gpt-3.5-turbo       | 0.465 |     0.684 |  0.552 | 0.611 |
| gpt-4o              | 0.544 |     0.675 |  0.650 | 0.662 |
| gpt-4-turbo         | 0.571 |     0.724 |  0.662 | 0.692 |
| llama-3-8b-instruct | 0.412 |     0.622 |  0.508 | 0.559 |
| llama-3-70b-instruct| 0.512 |     0.710 |  0.599 | 0.650 |
| REBEL               | 0.010 |     0.080 |  0.018 | 0.029 |
| IMoJIE              | 0.192 |     0.402 |  0.273 | 0.325 |

We find that larger and newer LLMs perform slightly better than others but they all outperform Open and Closed IE methods by large margins. This is due to IMoJIE usually extracting large portions of the passages being processed and being unable to do coreference resolution. On the other hand, ClosedIE models like REBEL fail dramatically because they extract very few entities and relations. Additionally, we refer the reviewer to this work which shows further evidence that LLMs can compete with supervised OpenIE methods in their training setting without further training [4].

We will add this important intrinsic evaluation to the camera ready version of our paper.

- **(R3-W1): Extracting triplets from the passage strongly depends on the power of the triplets extracting model, which is a language model in the paper's implementation. I'm concerned that it may lose information when the passage becomes longer. Is this a common strategy in other retrieval methods?**

We first point out that this information extraction methodology has in fact become more common recently [5, 6], however, it has not been directly applied to retrieval until now.

Second, we present both intrinsic and extrinsic experiments that help us understand the robustness of our OpenIE methods to passage length.

In our intrinsic length-dependent evaluation below, we present the gpt-3.5-turbo OpenIE results on the 10 shortest passages vs the 10 longest passages and find a substantial deterioration of OpenIE performance when extracting from longer passages.

Data Subset        | AUC   | Precision | Recall | F1
-------------------|-------|-----------|--------|-------
10 Shortest Docs   | 0.589 | 0.792     | 0.657  | 0.718
10 Longest Docs    | 0.390 | 0.607     | 0.485  | 0.539

In our extrinsic evaluation, we leverage the fact that the MuSiQue dataset contains several passages with the same Wikipedia article title. We therefore combine all passages with the same title into one document, creating longer documents for the OpenIE and DPR models to process. We find that our method outperforms Contriever by similar margins that in our standard setting even though the intrinsic knowledge graph quality is likely diminished. This result is indicative that HippoRAG’s joint components make it more robust than DPR with regards to longer passages.

#### Original MuSiQue:

Model      | R@2  | R@5
-----------|------|------
Contriever | 34.8%| 46.6%
HippoRAG   | 41.0%| 52.1%

#### Longer Document MuSiQue:

Model      | R@2  | R@5
-----------|------|------
Contriever | 32.6%| 41.7%
HippoRAG   | 45.7%| 57.7%

We note that the above original results are not directly comparable to the longer document results given that the number of total documents available for retrieval corpus changed. We include the dataset statistics for reference below:

Statistic | Original MuSiQue | Longer Document MuSiQue
----------|------------------|------------------------
Count     | 11,656           | 9,838
Mean      | 79.80            | 94.54
Std       | 47.93            | 146.01

We will add these length-dependent experiments to our updated paper.

### References

[1] Bhardwaj et al. (2019). CaRB: A Crowdsourced Benchmark for Open IE.\
[2] Huguet Cabot et al. (2021). REBEL: Relation Extraction By End-to-end Language generation.\
[3] Kolluru et al. (2020). IMoJIE: Iterative Memory-Based Joint Open Information Extraction.\
[4] Ling et al. (2023). Improving Open Information Extraction with Large Language Models: A Study on Demonstration Uncertainty.\
[5] Park et al. (2023). Graph Elicitation for Guiding Multi-Step Reasoning in Large Language Models.\
[6] Li et al. (2023). Leveraging Structured Information for Explainable Multi-hop Question Answering and Reasoning.

---

### Decision · Program_Chairs · 2024-09-25

**Decision:**

Accept (poster)

**Comment:**

This paper presents HippoRAG, a new "retrieval framework inspired by the hippocampal indexing theory of human long-term memory to enable deeper and more efficient knowledge integration over new experiences". HippoRAG utilises LLMs, knowledge graphs, and PageRank to mimic the different roles of neocortex and hippocampus in human memory. The authors experimentally demonstrate the effectiveness of their approach on multi-hop question answering and show substantial improvements over the state of the art. The authors have addressed the reviewers' concerns in the rebuttal and have included additional experiments as needed. The paper is well-written,  technically solid, with thorough and sound evaluation experiments. The idea is new and will serve as a solid foundation towards long-term memory solutions in LLMs.